# Vitamin D status modulates mitochondrial oxidative capacities in skeletal muscle: role in sarcopenia

Jérôme Salles[1], Audrey Chanet[1], Christelle Guillet[1], Anouk MM. Vaes[2], Elske M. Brouwer-Brolsma[2], Christophe Rocher [3], Christophe Giraudet[1], Véronique Patrac[1], Emmanuelle Meugnier[4], Christophe Montaurier[1], Philippe Denis[1], Olivier Le Bacquer[1], Adeline Blot[5], Marion Jourdan[6], Yvette Luiking[6], Matthew Furber[6], Miriam Van Dijk[6], Nicolas Tardif[7], Y. Yves Boirie[1,8] & Stéphane Walrand [1,8 ✉]

Skeletal muscle mitochondrial function is the biggest component of whole-body energy output. Mitochondrial energy production during exercise is impaired in vitamin D-deficient subjects. In cultured myotubes, loss of vitamin D receptor (VDR) function decreases mitochondrial respiration rate and ATP production from oxidative phosphorylation. We aimed to examine the effects of vitamin D deficiency and supplementation on whole-body energy expenditure and muscle mitochondrial function in old rats, old mice, and human subjects. To gain further insight into the mechanisms involved, we used C2C12 and human muscle cells and transgenic mice with muscle-specific VDR tamoxifen-inducible deficiency. We observed that in vivo and in vitro vitamin D fluctuations changed mitochondrial biogenesis and oxidative activity in skeletal muscle. Vitamin D supplementation initiated in older people improved muscle mass and strength. We hypothesize that vitamin D supplementation is likely to help prevent not only sarcopenia but also sarcopenic obesity in vitamin D-deficient subjects.

[1] Université Clermont Auvergne, INRAE, UNH, CRNH Auvergne, 63000 Clermont-Ferrand, France. [2] Wageningen University, Human Nutrition, Wageningen, the Netherlands. [3] Laboratoire de Biogenèse Membranaire - UMR 5200 CNRS, Université de Bordeaux, 33140 Villenave d'Ornon, France. [4] Univ Lyon, CarMeN Laboratory, INSERM, INRAE, INSA Lyon, Université Claude Bernard Lyon 1, 69310 Pierre-Bénite, France. [5] CHU Clermont-Ferrand, Centre de Recherche en Nutrition Humaine Auvergne, 63000 Clermont-Ferrand, France. [6] Specialized Nutrition, Danone Nutricia Research, P.O. Box 80141, 3584 CT Utrecht, the Netherlands. [7] Division of Perioperative Medicine and Intensive Care, Karolinska University Hospital, Huddinge, Sweden. [8] CHU Clermont-Ferrand, Service Nutrition Clinique, 63000 Clermont-Ferrand, France. ✉email: stephane.walrand@inrae.fr

Vitamin D status declines in people aged 70 years and older due to decreased exposure to sunlight and cutaneous synthesis. Vitamin D status is especially poor in institutionalized patients, 75% of whom are severely vitamin D-deficient, and in patients with hip fracture[1]. Vitamin D deficiency has been a focus of clinicians and researchers for decades due to its potential implications for global health, such as sarcopenia, osteoporosis, falls, fractures, impaired immune status, and chronic disease[2]. On a pre-clinical level, Bischoff-Ferrari et al. reported a direct association declining specific vitamin D receptor (VDR) expression in muscle cells and age-related loss of muscle mass and function[3]. Moreover, vitamin D has been suggested to exert pleiotropic effects during aging, with accumulating evidence linking low levels of vitamin D to body fat accumulation and obesity[4]. An adequate serum vitamin D level in mid- and late-life, i.e., serum 25-hydroxyvitamin D (25(OH)D) over 50 nmol/L or 20 ng/mL, was associated with reduced odds of multiple adverse body composition, especially sarcopenic obesity, suggesting potential health benefits of maintaining adequate levels of vitamin D[5].

Body composition is dependent on body energy homeostasis, i.e., energy intake and total energy expenditure. Total energy expenditure is the sum of the basal metabolic rate (the amount of energy expended while at complete rest), the thermic effect of food (the energy required to digest and absorb food), and the energy expended in physical activity. Energy expenditure is thought to be mediated (at least in part) by vitamin D homeostasis. Hanks et al. found that single nucleotide polymorphisms in the VDR were associated with resting energy expenditure (REE)[6]. Furthermore, vitamin D status is a key regulator of the energy expended from a meal, i.e., thermic effect of a meal[7]. The biggest component of whole-body energy output is skeletal muscle oxidative activity, i.e., skeletal muscle mitochondrial function. Rana et al. and Sinha et al. were pioneers in reporting a beneficial effect of vitamin D supplementation on muscle weakness in severely vitamin-D-deficient but otherwise healthy adults[8,9]. This effect was explained by improved mitochondrial function, as measured in vivo using phosphorus-31 nuclear magnetic resonance spectroscopy. Moreover, the energy production by muscle mitochondria during the recovery phase after modest exercise is impaired in subjects with severe vitamin D deficiency. This slower energy generation in skeletal muscle mitochondria may play a role in decreased muscle strength and altered muscle metabolic homeostasis. In the same way, aging is associated with a reduced rate of energy production in muscle cells, likely associated with the age-related loss of muscle mass and contractile function[10].

Researchers have shown through the use of muscle cells in culture that the active form of vitamin D, i.e., 1,25-dihydroxyvitamin D (1,25(OH)$_2$D), could be a key regulator of muscle mitochondrial activity[11,12]. In cultured C2C12 myoblasts and myotubes, the loss of VDR function results in significantly lower mitochondrial respiration rates. In this study, the observed reductions in mitochondrial function were a result of a reduced ATP production from oxidative phosphorylation[12]. These findings suggest that the VDR plays a fundamental regulatory role in skeletal muscle mitochondrial function. Mitochondrial oxygen consumption rate was also increased in human skeletal muscle cells treated with vitamin D[11]. Respiration coupled to the generation of ATP was increased, which suggests that vitamin D increases energy production in muscle. Furthermore, a profound change in the expression of several hundred nuclear mRNAs, several of which encode mitochondrial proteins, was observed after treating cultured human muscle cells with the active vitamin D metabolite, i.e., 1,25(OH)$_2$D[11]. Note that the increase in mitochondrial function was specific for 1,25(OH)$_2$ vitamin D and did not occur with other vitamin D analogs that lack either one of both C-1 and C-25 hydroxyls. This observation is consistent with the high binding affinity of 1,25(OH)$_2$ vitamin D for the VDR relative to the lower affinities of the other analogs lacking hydroxyls[13].

The regulatory role of the VDR in skeletal muscle mitochondrial function remains largely underexplored. The aim of this study was to examine the effects of vitamin D deficiency and supplementation on whole-body energy expenditure and muscle mitochondrial oxidative function in old rats, old mice and human older subjects with or without vitamin D depletion. To gain further insight into the mechanisms explaining the action of vitamin D on muscle mitochondrial functions, we also used cultured murine and human muscle cells and transgenic mice with muscle-specific VDR tamoxifen-inducible deficiency. We hypothesized that suboptimal mitochondrial function contributes to the myopathy that onsets in vitamin D-deficient older individuals. First, we tested the impact of vitamin D depletion on whole-body energy expenditure, body composition and muscle mitochondrial function in old sarcopenic rats and human subjects. Second, we used microarray analysis in C2C12 and in muscles collected from old control and vitamin D-depleted rats to measure the effect of vitamin D on mitochondrial oxidative capacities and overall gene expression patterns. Third, we determined the effect of vitamin D supplementation on changes in muscle mass and function in vitamin D-deficient old mice and older human subjects. Fourth, we generated human skeletal actin-MCM-VDRfl/fl transgenic mice, i.e., transgenic mice with a skeletal muscle tissue-specific tamoxifen-inducible VDR deficiency, to evaluate the specific effect of the VDR on mitochondrial biogenesis and function in skeletal muscle.

## Results

**Long-term vitamin D deficiency modifies body composition of old rats.** Feeding old Wistar rats on a vitamin D-depleted diet for 9 months resulted in a significant decrease in plasma 25(OH)D concentrations compared to age-matched control animals ($-75\%$; $p > 0.0001$), reaching a severe state of vitamin D deficiency (i.e., <25 nM) (Fig. 1a) without affecting plasma levels of calcium, phosphorus and PTH (Supplementary Table 1). No significant difference in food intake was observed between the experimental groups throughout the study period (control rats $20.7 \pm 0.7$ g/day vs. vitamin D-depleted rats $21.5 \pm 0.7$ g/day). At the beginning of the experiment, there was no significant between-group difference in body weight (Fig. 1b). After 3 months, body weight gain was greater in vitamin D-depleted rats than in control rats ($+108\%$, $p < 0.05$), and this significant weight difference was maintained for 7 months. Thereafter, body weight decreased in both groups, but body weight gains were higher in vitamin D-depleted rats than controls (Fig. 1b).

After 9 months of the experiment, the control rats and vitamin D-depleted rats lost lean body mass with no significant between-group difference (Fig. 1c). Similar lean soft tissue weights (gastrocnemius, quadriceps, liver, and heart) were also observed in both groups (Supplementary Table 2). Note that the weight of plantaris muscle, a type II fiber-prominent muscle, tended to be lower in vitamin D-depleted rats than in control rats ($-18\%$, $p = 0.05$) (Fig. 1d). Both rat groups gained fat mass throughout the study period, but fat mass gain was significantly stronger in vitamin D-depleted rats than in control rats (Fig. 1e). More specifically, perirenal adipose tissue and subcutaneous adipose tissue weights were significantly higher in vitamin D-depleted rats than in control rats ($+60\%$ and $+82\%$, respectively) (Fig. 1f, g).

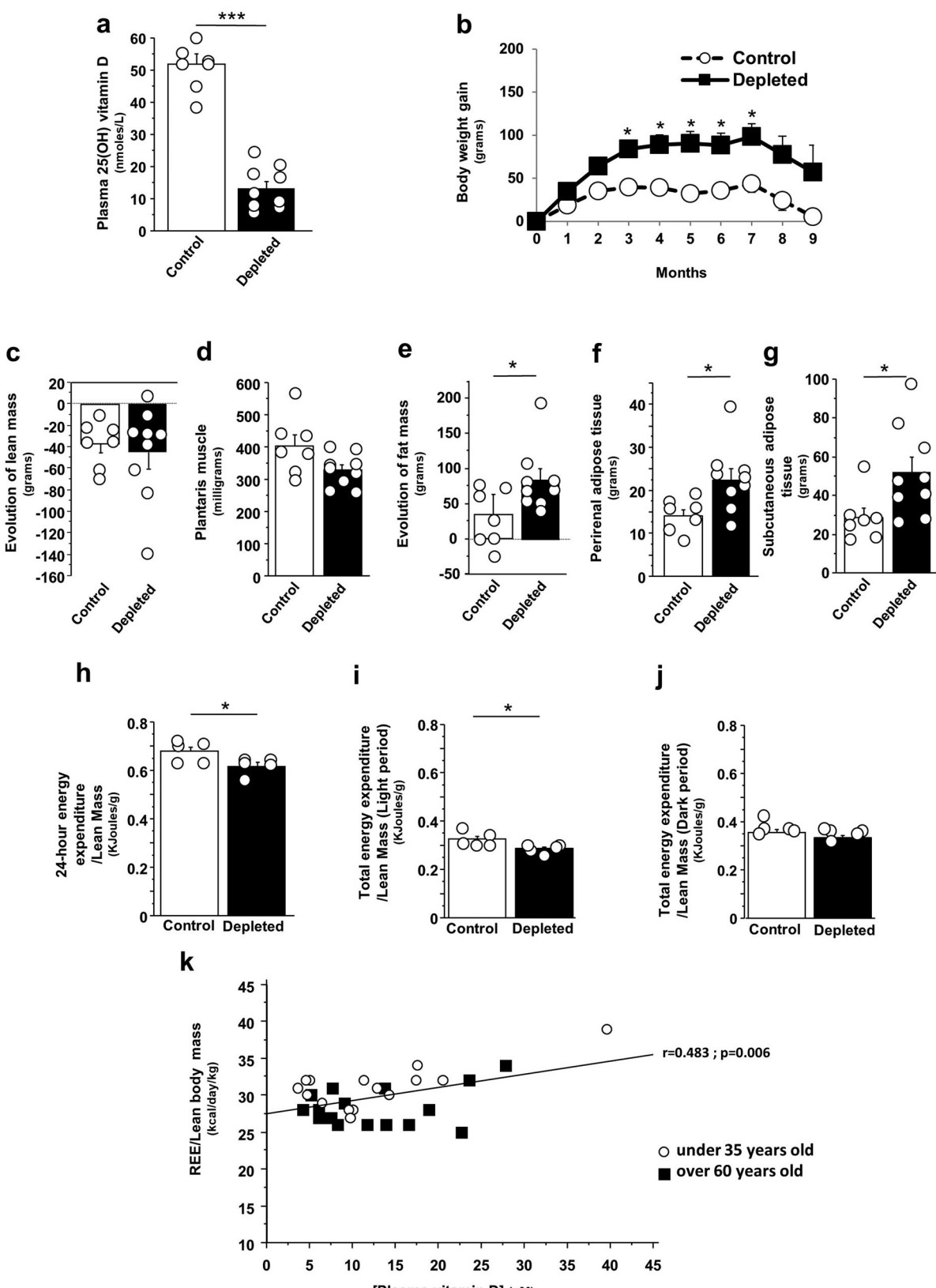

**Vitamin D plasma level associates with whole-body energy expenditure in rats and humans**. Given that perirenal and subcutaneous adipose tissues were heavier in vitamin D-depleted rats than in control rats, we investigated the mechanisms that could lead to this adipose tissue development. From an energy perspective, adipose tissue expansion is the result of an imbalance between caloric intake and energy expenditure. To further investigate the mechanisms leading to adipose tissue development with vitamin D deficiency, as food and energy intakes were similar between groups throughout the 9-month study, we analyzed whole-body energy expenditure using indirect calorimetry. Long-term vitamin D depletion significantly reduced 24-h energy

**Fig. 1 Vitamin D deficiency increases body weight and fat mass and influences whole-body energy expenditure. a** Plasma 25(OH)D concentrations in vitamin D-depleted old rats ($n = 9$) and in control old rats ($n = 7$) at the end of the 9-month vitamin D depletion period. **b** Time-course of body weight in vitamin D-depleted old rats ($n = 9$) and control old rats ($n = 7$) during the 9-month vitamin D depletion period. Mean changes in lean body mass (**c**) and fat body mass (**e**) from baseline to 9 months of vitamin D depletion in vitamin D-depleted old rats ($n = 9$) and in control old rats ($n = 7$). Weights of plantaris muscle (**d**), perirenal adipose tissue (**f**), and subcutaneous adipose tissue (**g**) in vitamin D-depleted old rats ($n = 9$) and control old rats ($n = 7$) at the end of the 9-month vitamin D depletion period. 24-h energy expenditure (**h**), energy expenditure during the light period (**i**), and energy expenditure during the dark period (**j**) adjusted to lean body mass in vitamin D-depleted old rats ($n = 5$) and control old rats ($n = 5$). **k** Correlations between resting energy expenditure (REE) adjusted to lean body mass and plasma 25(OH)D concentrations in 31 male subjects including 15 young (from 20 to 35 years old) and 16 older individuals (over 60 years old) ($r$ = Pearson's correlation coefficient). Data are expressed as means ± SEM. Differences between groups were analyzed with an unpaired $t$-test. *statistically different from control rats at $p < 0.05$. **statistically different from control rats at $p < 0.01$. ***statistically different from control rats at $p < 0.001$.

expenditure corrected for lean body mass in old rats (Fig. 1h) whereas ambulatory activity and energy intake were similar for both groups during this period (Supplementary Fig. 1a, d). As expected, energy expenditure was different between the light *vs.* dark phases in accordance with the inactive/resting and the active/feeding periods, respectively. Data obtained over the 24-h period, showed that energy expenditure was significantly lower during the light period and tended to be lower during the dark period in vitamin D-depleted rats than in their control counterparts (Fig. 1i, j). The two groups did not differ in energy intake and ambulatory activity during light and dark periods (Supplementary Fig. 1b, c, e, f). Taken together, the results show that the increase in adipose tissue due to vitamin D deficiency was primarily driven by reduced 24-h whole body energy expenditure and light-period energy expenditure. To confirm these data, we measured REE and plasma 25(OH)D concentrations in a group of 31 healthy men including 15 young (from 20 to 35 years old) and 16 older individuals (over 60 years old). Consistent with the data obtained in old rats, we found a significant and positive correlation between REE adjusted to lean body mass and plasma vitamin D concentration ($r = 0.483$; $p = 0.006$) (Fig. 1k).

Taken together, our data collected in rats and a group of healthy men points to a tight relationship between plasma vitamin D levels and REE.

**Vitamin D status affects muscle mitochondrial function.** Skeletal muscle is one of the major contributors to REE, mainly due to its metabolic activity and its relative contribution in terms of body mass. To clarify how vitamin D deficiency affected energy expenditure, we investigated muscle mitochondrial function, which is one of the major cellular processes involved in the modulation of energy production. First, we performed functional analyses of the mitochondrial respiratory chain. Maximal mitochondrial respiration rate (state 3 respiration) was measured in permeabilized fibers of rat plantaris muscle with different substrates, i.e., pyruvate/malate (Complex I activity), succinate (Complex II activity) and ascorbate/TMPD (Complex IV activity) after the 9-month vitamin D-depleted period in old rats (Fig. 2a). Maximal oxygen consumption was significantly lower in muscle permeabilized fibers from vitamin D-depleted rats than from control rats, regardless of the substrate used. More specifically, the state 3 respiration rate was lower by 20%, 24%, and 30% with pyruvate/malate, succinate, and ascorbate/TMPD, respectively, in the vitamin D-depleted group compared with control rats (Fig. 2a). We then measured the maximal activity of the respiratory chain complexes (I–IV) and citrate synthase, a mitochondrial matrix enzyme, in plantaris muscle homogenates (Fig. 2b). Complex I, III, and IV had markedly reduced activities in plantaris muscle with vitamin D deficiency (−45%, $p < 0.01$; −31%, $p < 0.05$; −52%, $p < 0.01$, vs. control rats, respectively) whereas complex II and citrate synthase activities were unaffected (Fig. 2b).

To better clarify whether vitamin D level influences mitochondrial function, we performed a vitamin D supplementation experiment in differentiated cultured skeletal muscle cells by treating C2C12 myotubes and human primary myotubes with 0, 1, or 10 nM of 1,25(OH)2 vitamin D3 for 72 h. C2C12 myotubes exposed to 1 and 10 nM 1,25(OH)2 vitamin D3 showed strongly increased mitochondrial complex I–IV activities compared with control cells (Fig. 2c). Similarly, citrate synthase activity was significantly enhanced by 1,25(OH)2 vitamin D3 treatments at 1 and 10 nM in differentiated C2C12 cells (+29% and +37%, respectively) and in human primary myotubes (+19% and +21%, respectively) (Fig. 2c, d).

Taken together, these in vivo and in vitro data show that muscle mitochondrial function is modulated by vitamin D status.

**Vitamin D regulates muscle expressions of genes involved in mitochondrial biogenesis and function.** To investigate the molecular mechanisms involved in the modulation of muscle mitochondrial function depending on vitamin D status, we performed quantitative reverse transcription polymerase chain reaction (RT-PCR) analysis to estimate mitochondrial content, biogenesis, and dynamics in plantaris muscle from vitamin D-deficient and control rats. A long-term vitamin D deficiency tended to decrease mitochondrial DNA (mtDNA) content relative to nuclear DNA (nDNA) content in plantaris muscle from old rats ($p = 0.09$) (Fig. 3a). In line with this result, muscle of vitamin D-depleted rats showed markedly reduced gene expressions of several transcriptional factors and co-activators involved in mitochondrial biogenesis, i.e., *PGC1α*, *NRF1*, *NRF2*, *PPARα*, and *PPARβ* (Table 1). mRNA levels of *COXIV*, a subunit of the mitochondrial cytochrome c oxidase, *CPT1b*, a key enzyme of fatty acid oxidation, and *UCP3*, a mitochondrial uncoupling transmembrane protein, were also strongly decreased in plantaris muscles of vitamin D-depleted rats in comparison with control rats (−24%, −53%, and −61%, respectively) (Table 1). In addition, muscle mRNA levels of proteins involved in mitochondrial fusion, i.e., *MFN1* and *MFN2*, were significantly downregulated in the vitamin D-depleted group while gene expression of *FIS1* (Mitochondrial fission 1 protein), a protein involved in mitochondrial fission, was unaffected (Table 1).

To further demonstrate that the expression of genes involved in mitochondrial biogenesis and function is affected by the vitamin D status, we used vitamin D-deficient old mice, i.e., mice fed a vitamin D-depleted diet for 14 months (Mouse experiment 1) (Supplementary Table 3). Consistent with the rat data, vitamin D deficiency significantly reduced *NRF1* gene expression in mice gastrocnemius muscle (−34% vs. control mice) (Fig. 3B). In addition, we observed a significant decrease in mRNA levels of *TFAM* (Mitochondrial transcription factor A) and *NDUFB5*, a subunit of Complex I, in muscles of vitamin D-depleted mice in comparison with control mice (−19% and −20%, respectively) (Fig. 3b). In contrast, muscle *PGC1α*, *NRF2*, and *COXIV* gene

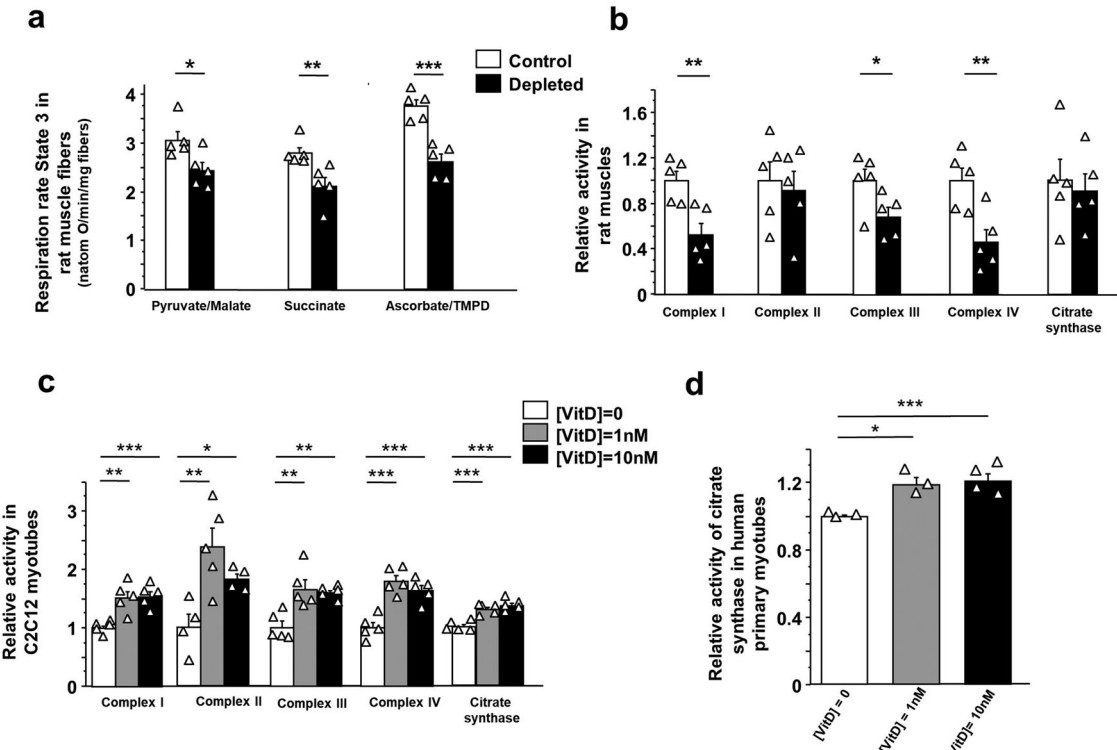

**Fig. 2 Vitamin D status modulates muscle mitochondrial function. a** Mitochondrial respiration in permeabilized fibers of plantaris muscles from vitamin D-depleted old rats ($n = 5$) and control old rats ($n = 5$). Respiration was expressed in natom O/min/mg of fibers and was measured at state 3 with: pyruvate (10 mM)/malate (10 mM), succinate (25 mM) in presence of rotenone (0.4 μg/ml), or ascorbate (3 mM)/N,N,N',N'-tetramethyl-p-phenylenediamine (TMPD) (0.5 mM). **b** Mitochondrial enzyme activities in plantaris muscle from vitamin D-depleted old rats ($n = 5$) and control old rats ($n = 5$), **c** in C2C12 myotubes treated with 0, 1, and 10 nM of 1,25(OH)2 vitamin D3, and **d** citrate synthase activity in human primary myotubes treated with 0 and 10 nM of 1,25(OH)2 vitamin D3 were expressed as fold change vs. control-group value. Cell culture data are combined from at least two independent experiments ($n = 3$–5). Data are expressed as means ± SEM. Differences between groups were analyzed with an unpaired $t$-test. *statistically different from the control group at $p < 0.05$. **statistically different from the control group at $p < 0.01$. ***statistically different from the control group at $p < 0.001$.

expressions were not affected by vitamin D deficiency in old mice (Fig. 3b). In line with the reduction of gene expression of *NDUFB5*, NDUFB8 protein, another subunit of the mitochondrial respiratory chain complex I, was less expressed in muscles of vitamin D-depleted mice (−66%, $p < 0.01$) (Fig. 3c). We found no significant differences for the other mitochondrial proteins whose content was measured (data not shown). Taken together, data from the mice and rat experiments suggest that long-term vitamin D deficiency decreases mitochondrial function and biogenesis through changes in key gene and protein expression.

To demonstrate a direct effect of vitamin D status on mitochondrial gene expressions, we also analyzed the mRNA level of proteins involved in mitochondrial biogenesis and function in C2C12 and human primary myotubes treated with 1,25(OH)2 vitamin D3. C2C12 myotubes treated with 10 nM of 1,25(OH)2 vitamin D3 exhibited a strong increase in levels of *PGC1α*, *PGC1β*, *NRF2*, *TFAM*, *PPARα*, *COXIV*, and *UCP3* transcripts compared to control cells (+83%, +123%, +40%, +68%, +39% and +297%, respectively) (Table 2). In addition, *FIS* mRNA was also overexpressed in vitamin D-treated C2C12 myotubes (+38% vs. control myotubes, $p < 0.01$) (Table 2). Consistent with these results, transcript levels of transcription factors and coactivators, *PGC1α*, *NRF2*, *TFAM*, and *PPARα*, and *CPT1b*, the rate-controlling enzyme of the fatty acid β-oxidation, were all significantly greater in human primary myotubes treated with 1,25(OH)2 vitamin D3 than in control cells (Fig. 3d). Based on gene expression analyses, western-blot quantifications showed a marked increase in mitochondrial

protein content in C2C12 cells treated with 1 and 10 nM of 1,25(OH)2 vitamin D3 (Fig. 3e, f). In addition, protein expression of Complex IV Subunit 4 tended to be higher after 1 nM 1,25(OH)2 vitamin D3 treatment ($p = 0.09$).

Taken together, our analyses in old rats and mice (Rat experiment and Mouse experiment 1) and in skeletal muscle cells suggest that vitamin D modulates muscle expression levels of genes involved in mitochondrial biogenesis and function in a coordinated way. Mitochondrial respiratory chain activity depends on the expression of hundreds of genes encoded in the nuclear and mitochondrial genomes. This implies that coordinated regulation of the cell's two genetic systems is necessary for a correct mitochondrial biogenesis. To explore this hypothesis, we performed a global analysis of genes that are differentially expressed (DEGs) in plantaris muscle of vitamin D-depleted old rats and in C2C12 myotubes according to vitamin D status. Microarray analyses showed that the levels of 1082 mRNAs were significantly changed in plantaris muscle after a 9-month period of vitamin D depletion in old rats. To understand the biological functions of the DEGs in plantaris muscle of vitamin D-depleted old rats, we conducted a functional analysis of all the genes that showed significant changes using the Database for Annotation, Visualization and Integrated Discovery (DAVID) to determine Gene Ontology (GO) term enrichments. Results indicated that DEGs were assigned to 34 significantly enriched GO terms. The top 10 enriched GO terms of the significantly upregulated DEGs are presented in Supplementary Fig. 2a. Upregulated DEGs were particularly involved in different GO terms related to myosin, muscle

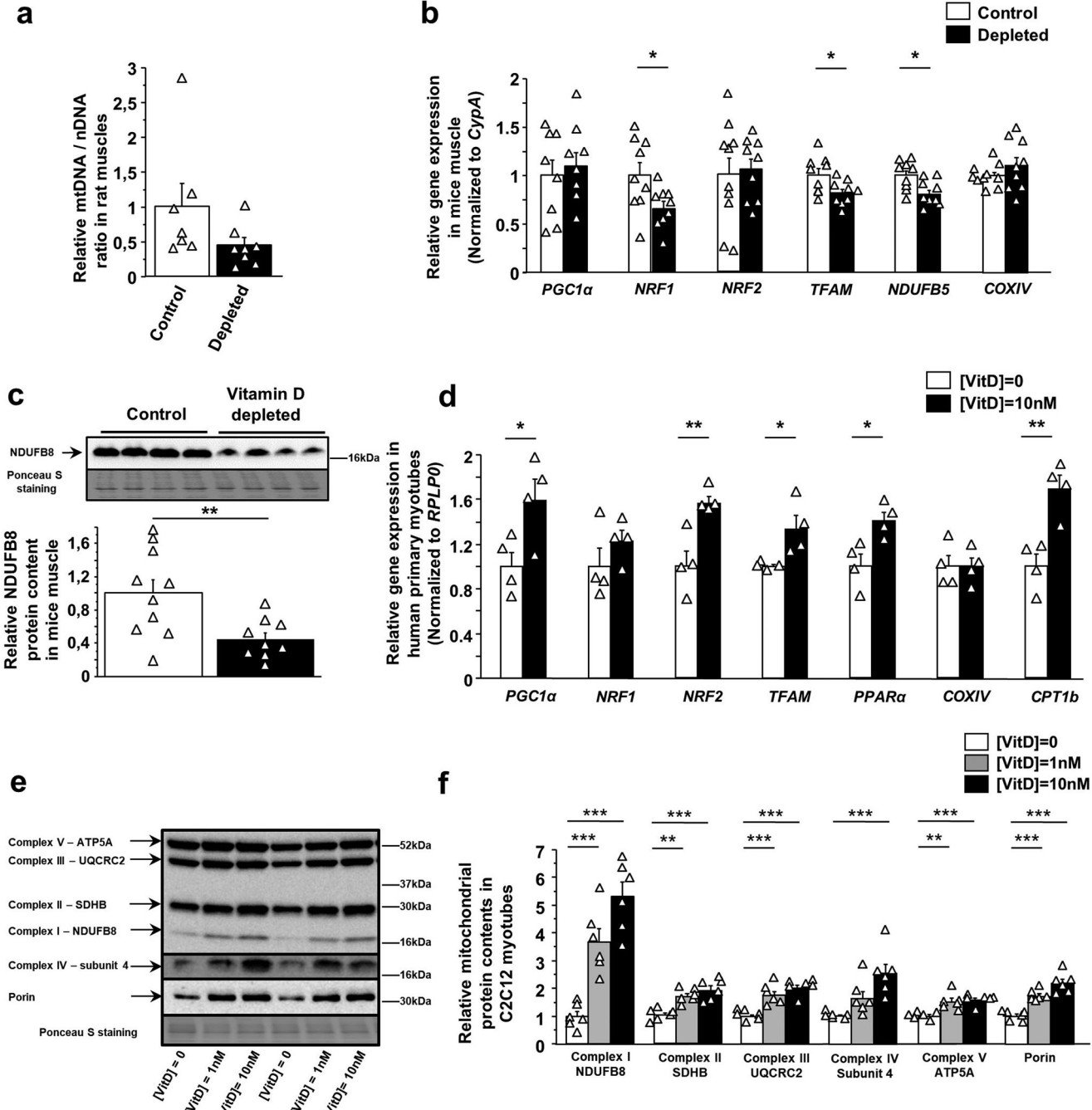

**Fig. 3 Vitamin D influences mitochondrial protein content and gene expression patterns in skeletal muscles, C2C12 and human primary myotubes.**
**a** mtDNA/nDNA ratio as a marker of mitochondria content in plantaris muscles from vitamin D-depleted old rats ($n = 8$) and control old rats ($n = 7$).
**b** Transcript levels of genes involved in mitochondrial biogenesis and function, and **c** protein content of NDUFB8, a respiratory chain complex I subunit, in gastrocnemius muscle from vitamin D-depleted old mice ($n = 9$) and control old mice ($n = 8$-10) (Mouse experiment 1). mRNA levels were normalized to the housekeeping gene *CypA*. **d** Transcript levels of genes involved in mitochondrial biogenesis and function in human primary myotubes treated with 0 and 10 nM of 1,25(OH)2 vitamin D3. mRNA levels were normalized to the housekeeping gene *RPLP0*. Representative immunoblots and red Ponceau staining (**e**), and immunoblot quantifications (**f**) of several subunits of each mitochondrial complex and Porin protein in C2C12 myotubes treated with 0 and 10 nM of 1,25(OH)2 vitamin D3. Cell culture data are combined from at least two independent experiments ($n = 4$-6). Data are expressed as means ± SEM and as fold change vs. control-group value. Differences between groups were analyzed with an unpaired *t*-test for rat and cell studies and with the Mann–Whitney *U*-test for mouse experiment 1. *statistically different from control animals or cells at $p < 0.05$. **statistically different from control animals or cells at $p < 0.01$, ***statistically different from control animals or cells at $p < 0.001$.

contraction, translation elongation, ribosomes, and cytosol (Supplementary Figure 2a). Functional enrichment analysis filtered with only the top 10 annotations for Gene Ontology BP, CC, KEGG, and Reactome components revealed that downregulated DEGs were enriched in annotations associated with purine metabolism, sarcoplasmic reticulum membrane, and mitochondria (Fig. 4a). In particular, DAVID analysis indicated that 51 downregulated DEGs were significantly enriched in a GO term related to mitochondria

**Table 1 Expression of genes involved in mitochondrial biogenesis, function, and dynamics in plantaris muscles from control ($n = 7$) and vitamin D-depleted ($n = 9$) rats.**

| Transcript levels | Control ($n = 7$) | Vitamin D-depleted ($n = 9$) | |
|---|---|---|---|
| *Mitochondrial biogenesis and function* | | | |
| PGC1α | 1.00 ± 0.16 | 0.50 ± 0.11 | * |
| PGC1β | 1.00 ± 0.11 | 0.68 ± 0.11 | |
| NRF1 | 1.00 ± 0.10 | 0.58 ± 0.07 | ** |
| NRF2 | 1.00 ± 0.10 | 0.58 ± 0.07 | ** |
| TFAM | 1.00 ± 0.03 | 0.87 ± 0.06 | |
| PPARα | 1.00 ± 0.04 | 0.63 ± 0.13 | * |
| PPARβ | 1.00 ± 0.14 | 0.45 ± 0.06 | ** |
| COXIV | 1.00 ± 0.07 | 0.76 ± 0.01 | * |
| CPT1b | 1.00 ± 0.07 | 0.47 ± 0.09 | *** |
| UCP3 | 1.00 ± 0.11 | 0.39 ± 0.09 | *** |
| *Mitochondrial dynamics* | | | |
| MFN1 | 1.00 ± 0.03 | 0.65 ± 0.09 | ** |
| MFN2 | 1.00 ± 0.09 | 0.42 ± 0.07 | *** |
| FIS1 | 1.00 ± 0.05 | 0.90 ± 0.07 | |

*RPLP0* gene expression was used to normalize all gene expression.
Data are expressed as means ± S.E.M. and as fold change vs. control-group value.
*statistically different from control rats at $p < 0.05$.
**statistically different from control rats at $p < 0.01$.
***statistically different from control rats at $p < 0.001$.

**Table 2 Expression of genes involved in mitochondrial biogenesis and dynamics in C2C12 myotubes treated with vehicle or 10 nM of 1,25(OH)$_2$ vitamin D$_3$ for 72 h.**

| Transcript levels | [VitD] = 0 | [VitD] = 10 nM | |
|---|---|---|---|
| *Mitochondrial biogenesis and function* | | | |
| PGC1α | 1.00 ± 0.11 | 1.83 ± 0.21 | ** |
| PGC1β | 1.00 ± 0.06 | 2.23 ± 0.19 | *** |
| NRF1 | 1.00 ± 0.06 | 1.18 ± 0.11 | |
| NRF2 | 1.00 ± 0.13 | 1.40 ± 0.12 | * |
| TFAM | 1.00 ± 0.04 | 1.17 ± 0.08 | |
| PPARα | 1.00 ± 0.11 | 1.68 ± 0.19 | * |
| PPARβ | 1.00 ± 0.08 | 1.04 ± 0.09 | |
| COXIV | 1.00 ± 0.06 | 1.39 ± 0.12 | * |
| CPT1b | 1.00 ± 0.09 | 0.96 ± 0.07 | |
| UCP3 | 1.00 ± 0.11 | 3.97 ± 0.50 | *** |
| *Mitochondrial dynamics* | | | |
| MFN1 | 1.00 ± 0.04 | 1.27 ± 0.13 | |
| MFN2 | 1.00 ± 0.09 | 1.14 ± 0.08 | |
| FIS1 | 1.00 ± 0.05 | 1.38 ± 0.08 | ** |

*RPLP0* gene expression was used to normalize all gene expression.
Data are expressed as means ± S.E.M. and as fold change vs. control-group value ([VitD] = 0) ($n = 6$ per condition).
*statistically different from control C2C12 myotubes $p < 0.05$.
**statistically different from control C2C12 myotubes at $p < 0.01$.
***statistically different from control C2C12 myotubes at $p < 0.001$.

(GO:0005739, mitochondrion) (Supplementary Table 4). Among these genes, we found 5 subunits of the respiratory chain complexes, 2 matrix mitochondrial enzymes, 4 mitochondrial ribosomal proteins, 1 mitochondrial translocase, 1 protein involved in mitochondrial dynamics, and the mitochondrial DNA polymerase (Supplementary Table 4). To complete these data, we performed a global transcriptomic analysis of DEGs in C2C12 myotubes treated with 10 nM of 1,25(OH)2 vitamin D3 in comparison with control C2C12 myotubes. The levels of 6068 mRNAs were significantly modulated by 1,25(OH)2 vitamin D3 treatment in skeletal muscle cells. DAVID analysis indicated that upregulated DEGs were assigned to 98 significantly enriched GO terms while downregulated DEGs were associated to 95 significantly enriched GO terms. Figure 4b presents the 'enrichment map' network created from the top 10 enriched GO terms (BP-CC-KEGG-REAC) of the upregulated DEGs in C2C12 myotubes treated with 10 nM of 1,25(OH)2 vitamin D3. Interestingly, enriched GO terms of the significantly upregulated DEGs in vitamin D-treated myotubes were mainly associated with respiratory electron transport, mitochondria organelle, and muscle contractile process. Downregulated DEGs were largely enriched in GO terms associated to collagen components and modifications, chromatin organization and cytoskeletal structure during cell division, and cholesterol biosynthesis (Supplementary Fig. 2b). Taken together, our global transcriptomic analyses with models of vitamin D deficiency and supplementation clearly indicate that vitamin D regulates genes involved in muscle mitochondrial function in a coordinated way.

**Vitamin D receptor is a key intermediate between vitamin D status and muscle mitochondrial function.** To further demonstrate that skeletal muscle is a direct target of vitamin D, we generated transgenic mice with muscle-specific VDR deficiency (mVDR KO mice; Mouse experiment 2). To induce the genomic recombination between loxP sites in the *VDR* gene, *HSA-MCM-VDR*fl/fl mice were orally gavaged with tamoxifen solution. A group of transgenic mice with the same genotype were orally gavaged with corn-oil vehicle and designated as controls. Tamoxifen administration significantly decreased *VDR* gene expression in skeletal muscles of *HSA-MCM-VDR*fl/fl mice (−75% in gastrocnemius muscle and −73% in tibialis muscle in comparison with vehicle-treated mice) (Fig. 5a, b). In contrast, *VDR* mRNA levels were not affected in other tissues such as liver, kidney, and small intestine (Supplementary Fig. 3a–c). Associated with the decrease in *VDR* expression, hindlimb muscle mass tended to be lower in mVDR KO mice ($p = 0.05$) (Fig. 5c).

To estimate mitochondrial function in muscles of mVDR KO mice, we performed maximal activity measurements of respiratory chain complexes, citrate synthase and 3-hydroxyacyl-CoA dehydrogenase (HAD), a key enzyme of fatty acid oxidation, in the gastrocnemius muscle (Fig. 5d). Specific muscle *VDR* deficiency induced a significant reduction of citrate synthase and HAD activity (−26% and −40%) and tended to decrease complex I, II, and IV activities ($p = 0.1$) while complex III activity appeared to be unaffected. In accordance with these data, muscle mitochondrial density evaluated using mitochondrial DNA-to-nuclear DNA ratio tended to be lower in mVDR KO mice than in controls ($p = 0.1$) (Fig. 5e). Furthermore, gene expressions of *NRF1* and *NRF2* were strongly downregulated in gastrocnemius muscle from mVDR KO mice (−54% and −61% vs. vehicle-treated mice, respectively; $p < 0.01$), and *PPARα* and *PPARβ* gene expression tended to decrease in muscles from mVDR KO mice ($p = 0.07$ and $p = 0.05$, respectively) (Fig. 5e). Interestingly, we found a strong positive correlation between *VDR* gene expression and both *NRF1* and *NRF2* mRNA levels in gastrocnemius muscles (Fig. 5f, g). Quantification of protein levels confirmed gene expression data and measurements of mitochondrial complex activities. Gastrocnemius muscle NRF1 and NRF2 protein contents were significantly lower in mVDR KO mice than in vehicle-treated mice (−20% and −35%, respectively). In addition, muscle *VDR* deficiency reduced muscle complex IV subunit 4 protein content (−22%, $p < 0.05$) (Fig. 5h, i). Taken together, our data from different animal and cell models definitively confirm that vitamin D regulates muscle mitochondrial function and biogenesis through highly coordinated modulation of gene expressions and that VDR is a key intermediate in this process.

**Vitamin D status and muscle function.** Given that mitochondria play a central role in muscle energy metabolism, we

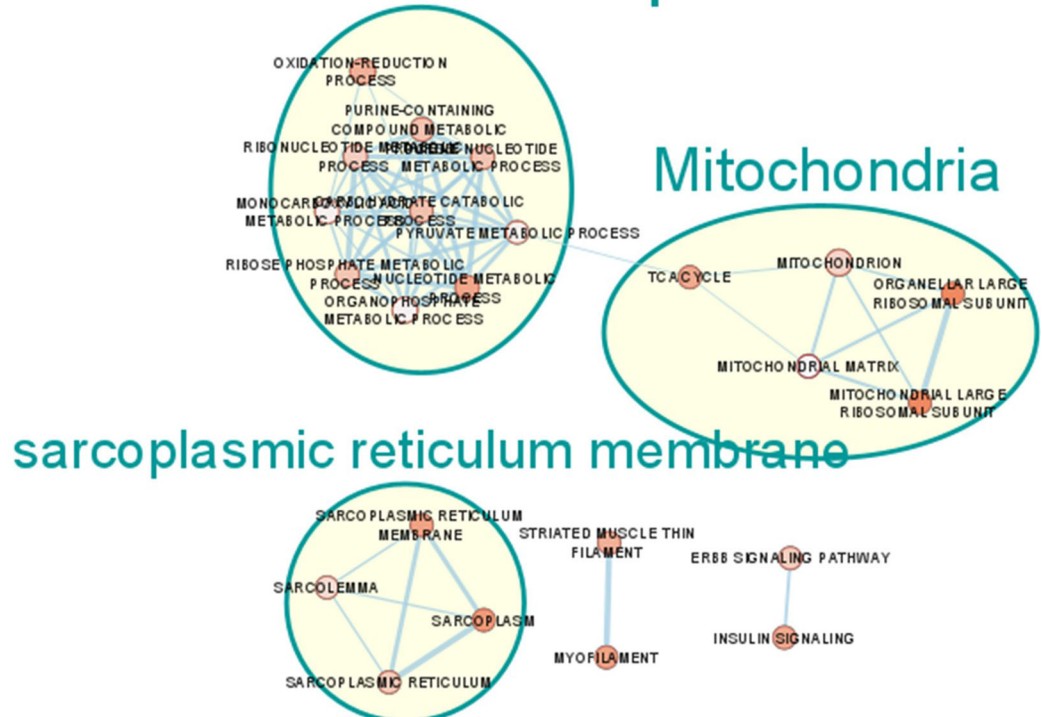

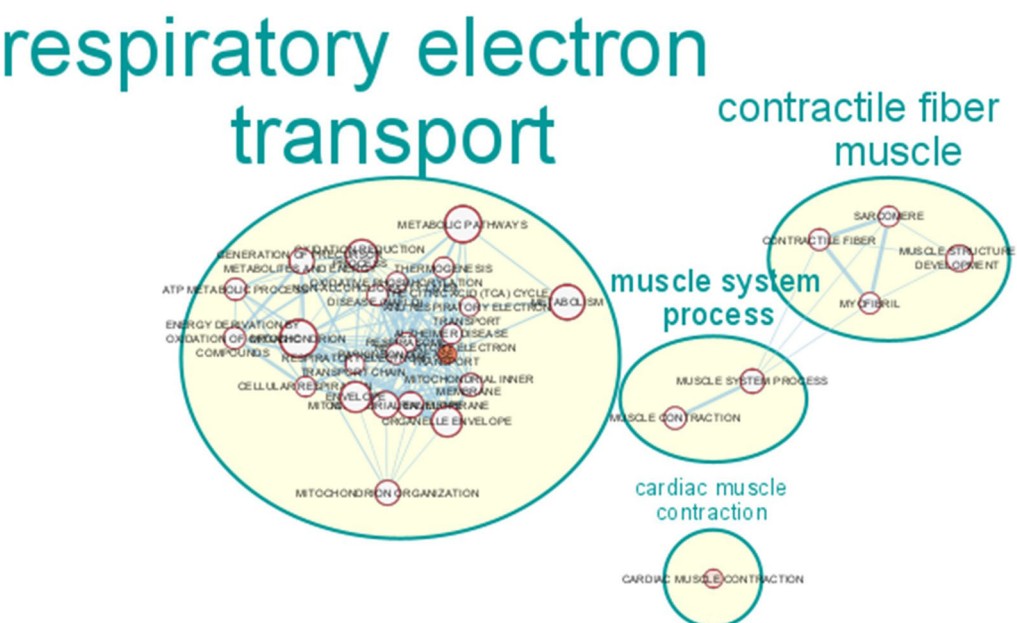

decided to explore muscle function according to vitamin D status. Vitamin D-deficient old mice showed significantly lower hindlimb muscle mass compared with their control counterparts (Mouse experiment 1) (Fig. 6a). The vitamin D-deficient group also showed lower forelimb grip strength (−21% vs. control mice, p < 0.05) (Fig. 6b). These data suggest that

vitamin D deficiency affects muscle mass and function in old mice. To confirm that vitamin D supplementation could also modulate these muscle parameters in older humans, we evaluated muscle mass and strength gain in a group of vitamin D-deficient elderly people subjected to a 6-month program of vitamin D supplementation. At the beginning of the

**Fig. 4 Vitamin D regulates expression levels of genes involved in mitochondrial function in old rat muscles and C2C12 cells in a coordinated way.**
DAVID was used for Gene Ontology (GO) enrichment analysis of significant differentially expressed genes (DEGs) in plantaris muscles of old rats ($p < 0.05$) and in C2C12 myotubes (B–H adjusted $p$-values < 0.05) according to vitamin D status. The top ten significant enriched GO terms of the DEGs downregulated in response to vitamin D deficiency in rat plantaris muscle (**a**) and the top ten significant enriched GO terms of the DEGs upregulated in response to vitamin D supplementation in C2C12 myotubes (**b**) are presented. Using the Cytoscape applications, EnrichmentMap and AutoAnnotate, related functional annotations (annotations of these functions are all in capitalized letters) were clusterised and an annotation of these clusters (lower case annotations) was added. The size of the cluster's annotation font relies on the size of the cluster generated (the higher the number of nodes is the bigger the font is).

supplementation, the appendicular skeletal muscle mass was significantly lower in the placebo group compared with the vitamin D-supplemented group ($16.8 \pm 0.5$ kg vs. $21.6 \pm 0.8$ kg, respectively), while no significant difference was observed for the grip strength ($20.6 \pm 0.7$ kg vs. $21.4 \pm 0.7$ kg, respectively). Subjects who received 10,000 IU of cholecalciferol three times per week for 6 months exhibited a significantly greater increase in plasma 25(OH) D from baseline compared with placebo-group participants (Fig. 6c). Furthermore, mean changes in appendicular skeletal muscle mass and handgrip strength from baseline to 6 months were very significantly higher in the vitamin D-supplemented group than in the placebo group ($+0.57 \pm 0.09$ kg vs. $+0.07 \pm 0.05$ kg for appendicular skeletal muscle mass gain and $+0.85 \pm 0.31$ kg vs. $+0.09 \pm 0.20$ kg for handgrip strength gain) (Fig. 6d, e). Finally, we found a significant positive correlation between plasma 25(OH)D concentration increase and appendicular skeletal muscle gain over the 6-month treatment period (Fig. 6f). All these results highlight a relationship between plasma vitamin D status and muscle function.

## Discussion

This study shows evidence of the existence of a direct vitamin D-mediated mitochondrial pathway in muscle and attempts to clarify the importance of the genomic driving of the mitochondrial activity by vitamin D using different experimental models. The unique findings include the fact that vitamin D depletion in old rats is associated with decreased whole-body energy expenditure related to a switch in body composition, i.e., increased fat mass and decreased skeletal muscle mass, and with compromised muscle mitochondrial activity. Although, it did not give any cause–effect relationship, a correlation between vitamin D fluctuation and metabolism and REE and muscle mass changes was observed in older human subjects and muscle-specific *VDR* KO mice. This work found convincing evidence that the effect of vitamin D occurs through a nuclear mechanism leading to a change in expression of the main regulators of mitochondrial biogenesis and function. Therefore, the involvement of a direct mitochondrial vitamin D pathway leading to rapid stimulation of mitochondrial oxidative activity has to be considered. A direct role for the VDR in regulating skeletal muscle mitochondrial respiration in vitro has been reported, providing a potential mechanism as to how vitamin D deficiency might impact upon skeletal muscle oxidative capacity[12]. In addition, the present study demonstrated a putative role of vitamin D depletion on key mitochondrial function parameters in skeletal muscle. Some previous data conducted in vitamin D deficiency in mice suggest that vitamin D-mediated regulation of mitochondrial function may underlie the exacerbated muscle fatigue and performance deficits observed during vitamin D deficiency[14]. The fact that vitamin D supplementation increased muscle mass and function in vitamin D-deficient older subjects highlights the many implications of vitamin D for clinical practice in sarcopenic subjects or for the prevention of sarcopenia and sarcopenic obesity.

The idea for this present study came from a previous observation—reconfirmed here—showing that vitamin D depletion caused a significant increase in body weight in rats due to a large accumulation of fat mass[15]. It has been reported consistently in the literature that overweight and obese individuals have lower 25(OH)D levels compared to non-obese individuals[16]. The underlying explanations and direction of causality of the increased risk of vitamin D deficiency in people who are overweight or obese are unclear[17]. Several hypotheses have been assessed. First, as vitamin D is stored in the adipose tissue, Wortsman et al. linked these lower levels to possible sequestration of vitamin D in the subcutaneous fat, i.e., obese individuals would have a larger vitamin D storage capacity[18]. However, although vitamin D is found in adipose tissue, it can only be released when the stored fatty acids are mobilized to supply energy[19,20]. Thus, sequestered vitamin D in adipocytes should not be regarded as a functional store. Second, active vitamin D (1,25-dihydroxyvitamin D) may influence the mobilization of free fatty acids from the adipose tissue, and thus explain why a deficiency can induce a gain of fat mass[21]. However, randomized controlled trials testing the effect of vitamin D supplementation on weight loss in obese or overweight individuals have provided inconsistent findings[22]. In these trials, the authors were giving one dose of vitamin D to all participants independently of age and body mass, which could have led to a less effective increase in 25(OH)D level, which in turn could have impacted the changes in body mass. Finally, in vivo experiments in rats have shown that large doses of vitamin D lead to increases in energy expenditure due to changes in oxidative phosphorylation rate in skeletal muscle and adipose tissues[11,23]. In the present work carried out in animals, the use of a dynamic model, i.e., vitamin D depletion while maintaining the same diet and the same food intakes as control animals, shows that the depletion of vitamin D could be the causal factor in weight gain in old rats and mice. We did not notice any fluctuation in plasma concentrations of calcium, phosphorus and PTH, so we can hypothesize that the observed effects are specific to vitamin D depletion. The effect of vitamin D fluctuations on body composition, i.e., the increase in fat mass, can be explained by its regulating role in energy homeostasis. We measured whole-body energy expenditure and mitochondrial activity in the tissue responsible for the body's greatest energy expenditure, i.e., skeletal muscle. We clearly showed in rodents and in men that vitamin D fluctuations are associated with the control of energy expenditure. Although our data did not show overt abnormalities of REE in vitamin D-deficient human subjects compared with healthy controls, the wide range of healthy control data reflects the broad biological variability in energy expenditure. The linear regression model therefore confirmed the relationship between vitamin D status and REE. REE decreases with decreasing vitamin D levels. It has also been reported that baseline 25(OH)D was positively correlated with both energy expended from a meal and fat oxidation in overweight women[7]. In addition, polymorphism of the *VDR* was associated with REE in children, and the mechanisms underlying these associations were influenced by some variables, e.g., adiposity[6]. Taken together, these results

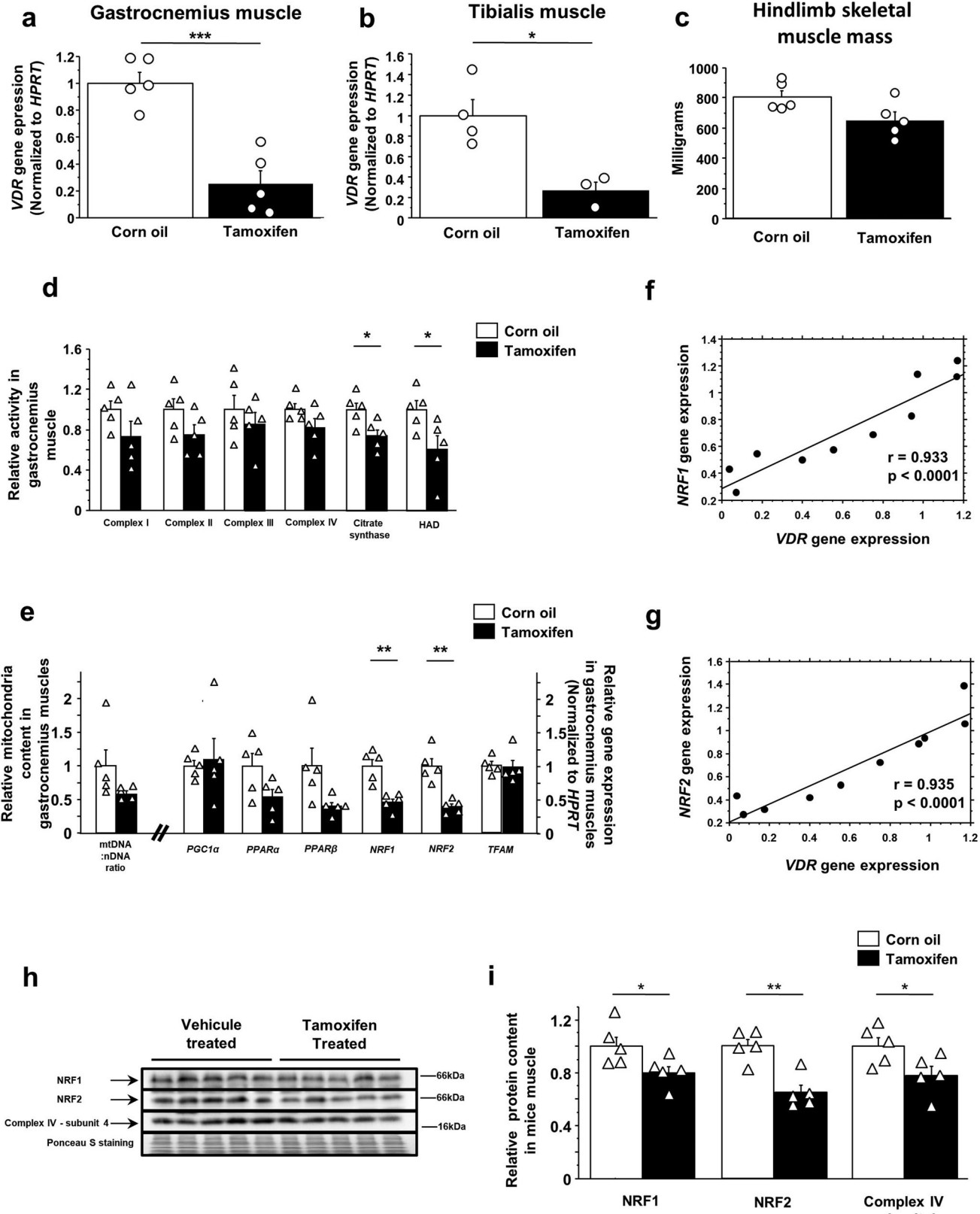

argue that vitamin D status may be related to energy expenditure and fat oxidation. The positive association between the vitamin D status and energy expenditure in older subjects and old rats, as observed in our study, could contribute to the increase in adipose tissue with age in vitamin D-deficient subjects.

We chose to focus on muscle mitochondrial activity because of its key role in explaining changes in whole-body energy expenditure[24]. Moreover, evidence suggests that, in addition to regulating skeletal muscle mass and function, vitamin D may also regulate skeletal muscle mitochondrial function[11,12]. These

**Fig. 5 Tamoxifen administration decreases muscle *VDR* gene expression resulting in a reduction of muscle mitochondrial biogenesis and function in *HSA-MCM-VDR*fl/fl transgenic mice.** *Vitamin D receptor* (*VDR*) transcript expression levels in **a** gastrocnemius and **b** tibialis muscles of tamoxifen-treated *HSA-MCM-VDR*fl/fl mice ($n = 4-5$) and corn oil vehicle-treated *HSA-MCM-VDR*fl/fl mice ($n = 3-5$) (Mouse experiment 2). mRNA levels were normalized to *HPRT*. Data are expressed as means ± SEM and as fold change vs. corn oil-treated group value. **c** Hindlimb skeletal muscle mass of tamoxifen-treated *HSA-MCM-VDR*fl/fl mice ($n = 5$) and corn oil vehicle-treated *HSA-MCM-VDR*fl/fl mice ($n = 5$) (Mouse experiment 2). **d** Mitochondrial enzyme activities, **e** mtDNA/nDNA ratio as marker of mitochondria content and transcript levels of genes involved in mitochondrial biogenesis and function in gastrocnemius muscle of tamoxifen-treated *HSA-MCM-VDR*fl/fl mice ($n = 5$) and corn oil vehicle-treated *HSA-MCM-VDR*fl/fl mice ($n = 5$) (Mouse experiment 2). Correlations between **f** NRF1 and *VDR* gene expressions, and **g** NRF2 and *VDR* gene expressions in tamoxifen-treated and corn oil-treated *HSA-MCM-VDR*fl/fl mice (Mouse experiment 2) ($r$ = Pearson's correlation coefficient). Immunoblots and red Ponceau staining (**h**), and immunoblot quantifications (**i**) of NRF1, NRF2, and complex IV subunit 4 proteins in gastrocnemius muscle of tamoxifen-treated *HSA-MCM-VDR*fl/fl mice ($n = 5$) and corn oil vehicle-treated *HSA-MCM-VDR*fl/fl mice ($n = 5$) (Mouse experiment 2). Data are expressed as means ± SEM. Differences between mouse groups were analyzed with an unpaired *t*-test. *statistically different from corn oil-treated mice at $p < 0.05$. **statistically different from corn oil-treated mice at $p < 0.01$. ***statistically different from corn oil-treated mice at $p < 0.001$.

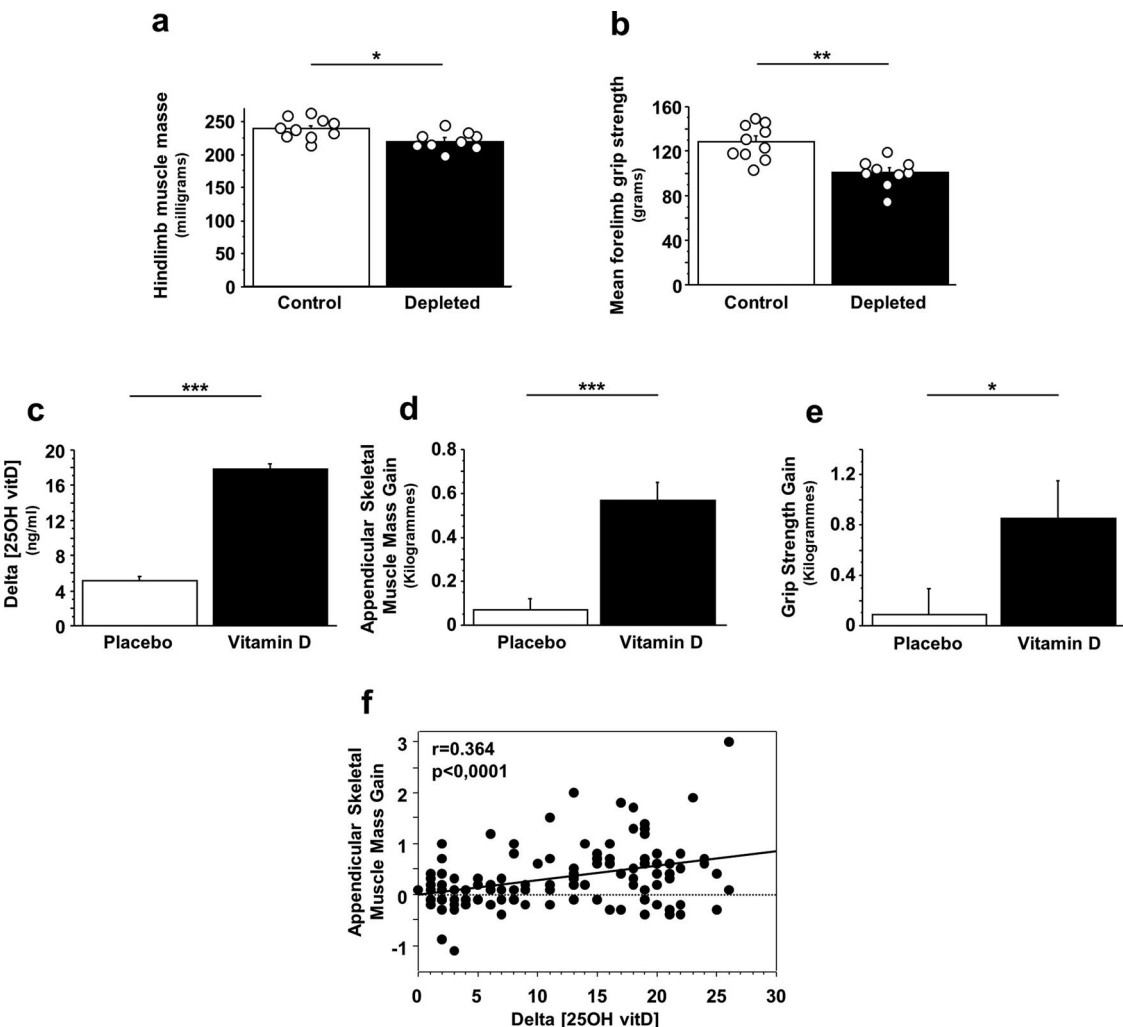

**Fig. 6 Vitamin D status modulates muscle mass and function in old mice and in vitamin D-deficient older subjects. a** Hindlimb mass and **b** forearm grip strength were measured in 24-month-old mice that received a vitamin D-depleted diet for 14 months (depleted; $n = 9$) and in 24-month-old mice that received a vitamin D-adequate diet for 14 months (control; $n = 10$) (Mouse experiment 1). Data are expressed as means ± SEM. Vitamin D-deficient older subjects ($n = 115$) received a supplement of 10,000 IU cholecalciferol three times per week for 6 months (33 men and 27 women) or a placebo (26 men and 29 women). Plasma 25(OH)D concentration, appendicular skeletal muscle mass and handgrip strength were measured at baseline and at 6 months. Changes in **c** plasma 25(OH)D concentration, **d** appendicular skeletal muscle mass, and **e** handgrip strength during the experimental period are presented. **f** Correlations between appendicular skeletal muscle mass gain and plasma 25(OH)D concentration increase (human trial) ($r$ = Pearson's correlation coefficient). Data are expressed as means ± SEM. Mouse experiment 1: Differences between groups were analyzed using the Mann–Whitney $U$ test. *statistically different from control mice at $p < 0.05$. Human trial: Differences between groups were analyzed using an unpaired *t*-test. *statistically different from placebo-treated subjects at $p < 0.05$. ***statistically different from placebo-treated subjects at $p < 0.001$.

observations are confirmed by our data using different cell and in vivo models. We show for the first time that vitamin D deficiency contributes to the aging-relation decline in muscle mitochondrial activity. Although a number of mechanisms have been proposed as the underlying causes of sarcopenia, mitochondrial abnormalities have been suggested as one of the key factors in muscle changes[10]. Research on the mitochondrial electron transport chain (ETC) in skeletal muscle has clearly demonstrated deficient ETC activity in the muscles that exhibit the greatest loss of mass with age[25]. A decreased state 3 respiration rate was observed in vitamin D-deficient animals. State 3 respiration rate was demonstrated by adding ADP to the mitochondrial preparation in the presence of suitable substrates. In state 3, ATP synthesis was increased while mitochondrial membrane potential was decreased, and the ETC consumed oxygen in order to restore membrane potential. The vitamin D depletion-induced decrease in state 3 in old rats highlighted abnormal ETC coupling between oxygen consumption and ADP phosphorylation rates in the mitochondria. In the same way, Ashcroft et al. recently reported that 3 months of diet-induced vitamin D deficiency reduced mitochondrial respiration and maximal ETC capacity in young C57BL/6 J mice[12]. Mitochondrial oxygen consumption is a function that reflects a composite of mechanisms including mitochondrial number, oxidative enzyme content and activity, mitochondrial components, and vascular supply of substrates and oxygen. Disruption to any or all of the above mechanisms may lead to suboptimal mitochondrial oxidative function. Here we observed a decrease in mitochondrial biogenesis and, at the mitochondrial level, changes in ETC complex activities. Inversely, vitamin D supplementation in different cell systems, i.e., C2C12 cells and human myotubes, dose-dependently increased the activity and protein expression of mitochondrial complexes and citrate synthase as well as mitochondrial mRNA transcript levels. In this regard, we show here using microarray analysis that at the muscle level, mitochondrial genes are among the genes most regulated by vitamin D, whether during supplementation of myotubes or after a period of depletion in old rats. In old vitamin D-depleted rats, we found a significant change in the expression of 51 genes encoding key proteins known to localize in mitochondria. Some of these data confirm the results obtained using qRT-PCR, in particular concerning the complexes of the respiratory chain in vitamin D-depleted old rats and in vitamin D-treated C2C12 cells.

Hormone-responsive elements on the mitochondrial genome could upregulate cellular energy production through genomic and nongenomic mechanisms[26]. Vitamin D receptors mediate the genomic action of 1,25 dihydroxyvitamin D and are expressed in several tissues. We therefore examined the effects of functional loss of VDR gene on mitochondrial activity and function by cross-breeding mice with flox sites flanking exon 2 of the VDR gene and mice containing a skeletal muscle tissue-specific tamoxifen-inducible Cre recombinase (MER-Cre-MER) to obtain HSA-MCM-VDRfl/fl transgenic mice. Male and female adult HSA-MCM-VDRfl/fl mice received oral gavage of tamoxifen to inactivate the VDR gene specifically in skeletal muscles. All the resultant muscle-specific VDR KO mice developed abnormal muscle mitochondrial function and regulation. The newly created muscle-specific VDR KO mice showed alterations in mitochondrial biogenesis and in transcript and protein levels of oxidative enzymes and transcription factors in skeletal muscles but no change in other tissues. This result clearly shows that vitamin D exerts its action on muscle mitochondrial functions at least partly through the activity of its receptor. To examine the regulatory role of the VDR in skeletal muscle mitochondrial function, Ashcroft et al. utilized lentiviral-mediated shRNA silencing of the VDR in C2C12 myoblasts. They found that loss of VDR function

results in significant reductions in mitochondrial respiration from oxidative phosphorylation in both myoblasts and myotubes[12].

Vitamin D inadequacy or deficiency and muscle mitochondrial dysfunction are co-associated with sarcopenia in humans[10,27,28]. Here, we show that vitamin D deficiency blunts muscle mass and strength in old mice and that a 6-month vitamin D supplementation can improve muscle mass and strength in older vitamin D-deficient human subjects. We found a positive correlation between the degree of improvement in vitamin D status and vitamin D treatment-induced muscle mass gain. Although we could not measure mitochondrial oxidative capacity in these subjects, it is likely that muscle improvements were partly explained by an increased mitochondrial activity. Of note, we also showed that these subjects lose fat mass after the period of vitamin D supplementation, which fits with our hypothesis of a parallel increase in lipolytic capacities and mitochondrial activity induced by vitamin D[27,28].

This work has some technical limitations. The first refers to the 25(OH)D assay method. The technique used in this work, i.e., ELISA, as developed in recent years make it possible to obtain accurate and precise results. Nevertheless, the gold standard method for measuring the blood concentration of 25(OH)D remains mass spectrometry. Second, in the present study, body composition was measured in Humans by using the BIA technique. It has been shown that this technique is not the most reliable and accurate for defining skeletal muscle mass. It would have been better to use other techniques like dual-energy X-ray absorptiometry or magnetic resonance imaging, as we did in rodents. Hence, the results concerning the muscle mass measured in humans must be considered in our study bearing in mind that the measurement was carried out with the BIA technique.

In conclusion, this study tested the hypothesis that suboptimal mitochondrial function contributes to the myopathy found in vitamin D-deficient individuals and that vitamin D therapy is associated with improved primary mitochondrial function in skeletal muscle. We observed that in vivo and in vitro vitamin D fluctuations changed mitochondrial biogenesis and oxidative activity in skeletal muscle cells, notably in muscle-specific VDR KO mice. Our data indicates that vitamin D supplementation initiated in older people improves skeletal mass and strength. Taken together, our evidence points to a direct link between improved ATP availability in muscle cells, enhanced muscle mass and function, and increased vitamin D status in these subjects. This study also supports the hypothesis that vitamin D supplementation is likely to help prevent not only sarcopenia but also sarcopenic obesity in vitamin D-deficient subjects.

## Methods

**Human trials: impact of vitamin D status (observational study) and vitamin D supplementation (interventional study).** The observational study protocol was approved by the local Ethical Review Board of the General Clinical Research Center (GCRC) at Clermont-Ferrand, France, in accordance with the Declaration of Helsinki. Each participant gave written informed consent after being explained the purposes, methodology and potential risks of the study. This trial was registered at clinicaltrials.gov as NCT01066091. Thirty-one healthy male subjects, including 15 young (from 20–35 years old) and 16 older individuals (over 60 years old), with a body mass index (BMI) between 21.4 and 34.2 kg/m$^2$ were included in the study, as previously reported[29,30]. Each subject had a normal blood biochemical profile and was sedentary. Participants were not taking any medications liable to affect outcome parameters, i.e., corticosteroids, b-adrenergic blockers, lipid-lowering agents, or anticoagulants. On the day of the experiment, subjects were studied after an overnight fast of at least 10 h. Indirect calorimetry was performed when subjects were awake, in supine position in a quiet room (Deltatrac, Datex, Geneva, Switzerland). Gas samples were collected every minute for 1 h. The data of the first 10 min were not used because this period corresponds to the stabilization of the metabolism of the subjects under the canopy. The means of the $O_2$ consumption and $CO_2$ production values obtained over the last 50 min period were used in the analysis. REE was calculated using the Weir equation[31]. Blood samples were then collected in EDTA-coated tubes. Plasma was prepared and stored at −80 °C until 25(OH)D concentration measurement using an ELISA kit (Promokine, Heidelberg, Germany).

The interventional study protocol was approved by the Ethics Committee of the institutional review board at Saint Charles Hospital, Beirut, Lebanon, in accordance with the Declaration of Helsinki. This trial was registered at clinicaltrials.gov as NCT02942732. One hundred and fifteen elderly subjects (mean age: $73.31 \pm 2.05$ years) with vitamin deficiency ($25(OH)D < 20$ ng/mL as per Institute of Medicine (IOM) recommendations) who were pre-sarcopenic (i.e., skeletal muscle mass/height$^2$ = 7.26 kg/m$^2$ for males and 5.45 kg/m$^2$ for females) were asked to join a 6-month randomized, controlled, double-blind intervention study[27,28]. A written informed consent was signed by all the subjects which was approved by the Ethics Committee of the hospital's review board. Participants were randomized into two groups: a vitamin D group (33 men and 27 women) received a supplement of 10,000 IU vitamin D3, i.e., cholecalciferol (Euro-Pharm International, Canada) three times per week, and a placebo group (26 men and 29 women) who received three times per week a similar-size and similar-color tablet containing microcrystalline cellulose (66.3%), starch (33.2%), and magnesium stearate (0.5%), per serving. The vitamin D and the placebo were presented in the same container. The interventions were led for a period of 6 months. Biochemical analyses and muscle assessments were performed at baseline and at six months. Fasting blood concentrations of vitamin D [25(OH)D] were measured by radioimmunoassay (DiaSorin, Stillwater, MN). Appendicular skeletal muscle mass was determined from bioimpedance analysis measurements (Tanita BC-418 Segmental Body Composition Analyzer, IL) and was expressed in kg[27,28]. It should be noted that this method of measuring body composition is not the most accurate compared to magnetic resonance imaging or dual-energy X-ray absorptiometry. Handgrip strength was measured at 9:00 am in the dominant hand with a Martin vigorimeter (Martin; Elmed, Addison, IL), and the force was expressed in kg.

**Rat experiment: long-term vitamin D depletion**. All animal procedures were approved by the institution's (INRAE) animal welfare committee to comply to the Ethical license of the national competent authority, securing full compliance with EU Directive 2010/63/EU on the use of animals for scientific purposes. Twenty 15-month-old male rattus norvegicus strain Wistar rats were purchased from the Janvier breeding center (Le Genest-St-Isle, France). The rats were housed at one rat per cage in opaque regulatory-sized cages. Sawdust was used as bedding for the animals and was renewed once a week. Room temperature and humidity were regulated at approximately 21 °C and 50%, respectively. Rats were held on a reversed 12-h light/dark cycle. Lights were off from 6:00am till 6:00 pm. Fat and lean body mass (g) were determined non-invasively using an echoMRI system as previously described[15]. The rats were then randomly divided into two groups according to body weight, fat mass and lean mass, and assigned ($n = 10$ per group, for a total of 20 rats) to either the AIN-93 M maintenance diet containing 1 IU vitamin D3/g diet (control rats) or to a modified AIN-93 M diet with no vitamin D for 9 months (vitamin D-depleted rats) (TestDiet, MO). During the experiment, 4 rats died or were euthanized because they developed a tumor. The results presented above are based on analyses conducted with data of the rats without any signs of underlying illness. Dioxygen consumption (VO$_2$), carbon dioxide production (VCO$_2$), food and drink intakes, and daily activity of control and vitamin D-depleted rats were measured using a PhenoMaster/LabMaster four-cage TSE System (Bad Homburg, Germany) at the end of the 9-month experimental period. Rats were housed individually in cages for indirect calorimetry. Energy expenditure was calculated using Weir's equation(31). Food and drink consumption was measured using a weight sensor. Spontaneous activity was assessed using a three-dimensional meshing of light beams.

On the day of sacrifice, the remaining control ($n = 7$) and vitamin D-depleted ($n = 9$) rats were weighed and sacrificed by exsanguination under anesthesia by inhalation with a mix of isoflurane and oxygen. Blood was collected from the aorta artery. Liver, heart, adipose tissues and hindlimb skeletal muscles were removed, weighed, snap-frozen in liquid nitrogen, and stored at −80 °C until later analysis. Plasma 25(OH)-vitamin D concentration was measured by ELISA kit (Promokine, Heidelberg, Germany) according to the manufacturer's instructions. ELISA kit was also used to determine parathormone (PTH) (Immunotopics, San Clemente, CA, USA). Plasma concentrations of calcium and phosphorus were assessed using a Konelab 20 analyzer (Thermo Electron, Waltham, MA, USA).

**Mouse experiment 1: long-term vitamin D depletion**. Procedures were conducted according to national and institutional guidelines on the care and use of animals and were reviewed and approved by the local Committee for the Care and Use of Laboratory Animals at Wageningen University to comply with EU Directive 2010/63/EU on the use of animals for scientific purposes. Thirty male 5-month-old C57BL/6j RJ mice were purchased from Janvier Laboratories (Le-Genest-Saint-Isle, France). Mice were housed in Makrolon® type-II cages individually to avoid dominance among pairs and to monitor dietary intake and prevent large differences in body weight within groups. The cages were cleaned every two weeks and contained standard woodchip bedding material. Room temperature and humidity were regulated at approximately 21 °C and 50%, respectively. Mice were held on a reversed 12-h light/dark cycle. Lights were off from 6:00am till 6:00 pm. All mice received a control diet (AIN-93M diet) until 10 months of age, and were then randomized according to body weight and fasting glucose concentration and subsequently subdivided to either a group ($n = 15$) that received the control diet or

a group ($n = 15$) that received the same diet with no vitamin D for 14 months. Mice were weighed once every two weeks. Forearm grip strength was measured at 23 months of age, using a calibrated grip strength device (Panlab, Cornella, Spain). Body composition was measured by DEXA scan using a PIXImus imager (GE Lunar, Madison, WI). Six mice were removed during the study or showed pathologies at necropsy. The results presented below are based on the analyses conducted with data of the mice without any signs of underlying illness. At the end of the study, the remaining control ($n = 10$) and vitamin D-depleted ($n = 9$) mice were sacrificed under anesthesia by inhalation with a mix of isoflurane and oxygen, and blood was collected. Hindlimb skeletal muscles were removed, weighed, snap-frozen in liquid nitrogen, and stored at −80 °C until later analysis.

**Mouse experiment 2: generation of human skeletal actin-MCM-VDRfl/fl transgenic mice**. All animal procedures were approved by the institution's (INRAE) animal welfare committee to comply with the Ethical license of the national competent authority, securing full compliance with EU Directive 2010/63/EU on the use of animals for scientific purposes. Mice with flox sites flanking exon 2 of the VDR gene (referred to as VDRfl/fl mice, gifted by Dr Geert Carmeliet, Katholieke Universiteit Leuven, Leuven, Belgium) and mice containing a skeletal muscle tissue-specific tamoxifen-inducible Cre recombinase (MER-Cre-MER) (referred to as HSA-MCM mice, gifted by Dr. Karyn Esser, University of Kentucky, Lexington, KY) were backcrossed to C57Bl6/J background mice ten times[32,33]. Then, VDRfl/fl mice and HSA-MCM mice were bred to obtain HSA-MCM-VDRfl/fl transgenic mice. Ten adult HSA-MCM-VDRfl/fl mice received oral gavage of tamoxifen dissolved in corn oil (0.2 mg per gram of body weight) or vehicle solution for five consecutive days the first week, two consecutive days two weeks later, and two consecutive days each following month. After 15 weeks of tamoxifen or vehicle treatment, mice were fasted overnight and then euthanized under anesthesia by inhalation with a mix of isoflurane and oxygen. Liver, kidney, small intestine and hindlimb skeletal muscles were removed, weighed, snap-frozen in liquid nitrogen and stored at −80 °C until later analysis. The genotype of the mice was determined by genomic PCR at birth and confirmed at the beginning of the muscle analyses.

**Cell culture**. C2C12 myoblasts (ATCC, Manassas, VA) were grown to 80–90% confluence in DMEM supplemented with 10% fetal calf serum at 37 °C in a 5% CO$_2$-humidified atmosphere. Cells were then induced to differentiate into myotubes by switching to DMEM containing 2% heat-inactivated horse serum[15,34]. Human primary skeletal muscle myoblasts (Lot#: SLSK002) were purchased from Zenbio inc. (New York City, NY) and cultured following the company's instructions. After 5 days of differentiation, C2C12 myotubes or human skeletal myotubes were cultured for 3 days with 0, 1, or 10 nM of 1,25(OH)2 vitamin D3 dissolved in ethanol[15,34].

**Western blot analysis**. C2C12 myotubes and mice gastrocnemius muscles were homogenized in an ice-cold lysis buffer (50 mM HEPES pH 7.4, 150 mM NaCl, 10 mM EDTA, 10 mM NaPPi, 25 mM β-glycerophosphate, 100 mM NaF, 2 mM Na orthovanadate, 10% glycerol, 1% Triton X-100) containing a protease-inhibitor cocktail (1%) (Sigma # P8340) as we previously described[15,34,35]. Denatured proteins were separated by SDS-PAGE on 4–15% precast polyacrylamide gel (BIO-RAD, Marnes-la-Coquette, France) and transferred to a polyvinylidene membrane (Millipore, Molsheim, France). Immunoblots were incubated in a blocking buffer and then probed with primary antibodies: total OXPHOS Rodent WB Antibody Cocktail (Abcam, Paris, France), Complex IV subunit IV antibody (ThermoFisher Scientific, Courtaboeuf, France), NRF1 antibody (Genetex, Euromedex, Souffelweyersheim, France), NRF2 antibody (Genetex, Euromedex, Souffelweyersheim, France) or VDAC/Porin antibody (Biovision, Clinisciences, Nanterre, France). Immunoblots were then incubated with the corresponding horseradish peroxidase-conjugated secondary antibody (DAKO, Trappes, France). Luminescent visualization was done using ECL Western Blotting Substrate (Pierce, Thermo Fisher Scientific, Courtaboeuf, France) and an MF-ChemiBIS 2.0 imaging system (F.S.V.T., Courbevoie, France). The density of the bands was quantified using MultiGauge 3.2 software (Fujifilm Corporation). Ponceau S staining was used to normalize protein loading between samples. Data are expressed as fold change vs. control group value.

**RNA and DNA analysis**. Total RNAs and nuclear/mitochondrial DNA were extracted using Tri-Reagent (Euromedex, Mundolsheim, France) according to the manufacturer's instructions and as reported in our previous publication[35]. Total RNAs and mitochondrial and genomic DNA concentrations were quantified by measuring optical density at 260 nm[15,35]. Concentrations of mRNAs corresponding to genes of interest were measured by reverse transcription followed by PCR using a Rotor-Gene Q system (Qiagen, Courtaboeuf, France). Five micrograms of total RNA was reverse-transcribed using SuperScript III reverse transcriptase and a combination of random hexamer and oligo dT primers (Invitrogen, Life Technologies, Saint- Aubin, France). cDNAs were diluted 1:60 before PCR analysis. PCR amplification was performed in a 10 µL total reaction volume. The PCR mixture contained 2.5 µL diluted cDNA template, 5 µL 2× Rotor-Gene SYBR Green PCR master mix (Qiagen, Courtaboeuf, France), and 0.5 µM forward and reverse primers. The amplification profile was initiated by a 5-min incubation at 95 °C to activate HotStarTaq Plus DNA Polymerase, followed by 40 cycles of two steps: 95 °C for 5 s (denaturation step) and 60 °C for 10 s (annealing/extension step).

Relative mRNA concentrations were analyzed using Rotor-Gene software. Seven-fold serial dilutions from a mix of all undiluted cDNA were used for each target gene to construct linear standard curves from which the concentrations of the test sample were calculated. mRNA levels were normalized to the corresponding housekeeping genes depending on species and tissues as indicated in legends of Tables 1 and 2, Figs. 3 and 5, and Supplementary Fig. 3. Nuclear/mitochondrial DNA was isolated using Tri-Reagent followed by back extraction with 4 M guanidine thiocyanate, 50 mM sodium citrate, and 1 M Tris, and an alcohol precipitation as described previously by us[35]. Mitochondrial DNA (mtDNA) content was determined by quantitative real-time PCR analysis using a Rotor-Gene Q system (Qiagen, Courtaboeuf, France). To this end, the levels of *NADH dehydrogenase subunit 1* (*ND1*) (mitochondrial DNA) were normalized to the levels of *beta-actin* (genomic DNA). Data are expressed as fold change vs. control group value. The list of the primers used for real-time PCR amplification is reported in Supplementary Table 5.

**Mitochondrial enzymatic assays**. Fifty milligrams of frozen rat plantaris muscle, mice gastrocnemius muscle, C2C12 myotubes or human primary myotubes were homogenized with a glass–glass Potter in 9 volumes of homogenization buffer (225 mM mannitol, 75 mM sucrose, 10 mM Tris-HCl, 10 m EDTA, pH 7.2) and spun down at 650 g during 20 min at 4 ℃. The supernatant was kept and the pellet was suspended in 9 volumes of homogenization buffer and submitted to the same procedure. Both supernatants were pooled and used for the assay as previously described[35–38]. After protein quantification, activities of citrate synthase (CS), 3-hydroxyacyl-CoA dehydrogenase (HAD) and activities of complexes I–IV of the respiratory chain were spectrophotometrically assayed[35–39].

CS activity was measured by following, at 412 nm, the chemical reduction of 5,5-dithiobis(2-nitrobenzoicacid) (DTNB) by CoASH. The reaction mixture contained 200 mM Tris-HCl (pH 8.0), 300 μM acetyl-CoA (Sigma # A2181), 150 μM DTNB (Sigma # D8130), and 20 μg of supernatant proteins at 37 ℃. The reaction was initiated by adding 500 μM oxaloacetate (Sigma # O4126).

HAD activity was assessed by measuring the disappearance of NADH at 340 nm in presence of acetoacetyl-CoA. The reaction mixture contained 100 mM triethanolamine-HCl (pH 7.0), 5 mM EDTA, 120 μM acetoacetyl-CoA (Sigma # A1625) and 60 μg of supernatant proteins at 26 ℃. The reaction was initiated by adding 500 μM NADH (Sigma # N8129).

Complex I activity was measured by following the oxidation of NADH at 340 nm in presence of decylubiquinone. The reaction mixture contained 50 mM KH2PO4 (pH 7.5), 3.75 mg/ml bovine serum albumin (BSA) (Sigma # A2153), 75 μM decylubiquinone (Sigma # D7911), 0 or 5 μg/ml rotenone (Sigma # R8875) and 50 μg of supernatant proteins at 37 ℃. The reaction was initiated by adding 100 μM NADH (Sigma # N8129). Complex I activity was determined by calculating the difference between the activities measured in the absence and in the presence of rotenone.

Measurement of complex II activity was performed by following the decrease in absorbance at 600 nm resulting from the reduction of 2,6-dichlorophenolindophenol (DCPIP). The reaction mixture contained 50 mM KH2PO4 (pH 7.5), 1 mM potassium cyanide (KCN) (Sigma # 207810), 2 μg/mL rotenone (Sigma # R8875), 20 mM succinate (Sigma # S7501), 100 μM ATP (Sigma # A7699), 100 μM DCPIP (Sigma # 33125) and 40 μg supernatant proteins at 37 ℃. The reaction was initiated by adding 100 μM decylubiquinone (Sigma # D7911).

Complex III activity was determined after measuring, at 550 nm, ubiquinol cytochrome c oxidoreductase activity in the presence and in the absence of antimycin, a specific complex III inhibitor. The reaction mixture contained 80 mM KH2PO4 (pH 7.5), 2.5 mg/ml BSA (Sigma # A2153), 50 μM cytochrome c (Sigma # C7752), 1 mM potassium cyanide (KCN) (Sigma # 207810), 250 μM EDTA, 0 or 12.5 μg/ml antimycin A (Sigma # A8674) and 20 μg of supernatant proteins at 37 ℃. The reaction was initiated by adding 200 μM decylubiquinol. Decylubiquinol was prepared by mixing decylubiquinone (Sigma # D7911) and sodium borohydride (Sigma # 452882) in ethanol that leads to a fast decyluquiquinone reduction. The complex III activity is calculated by subtracting the antimycin insensitive activity from the total activity.

Complex IV activity (cytochrome c-oxidase activity) was determined by monitoring at 550 nm the oxidation of cytochrome c at 37 ℃. The reaction mixture contained 50 mM KH2PO4 (pH 7.0) and 100 μM reduced cytochrome c. The reaction was initiated by adding 20 μg of supernatant proteins. Reduced cytochrome c was prepared by adding sodium dithionite (Sigma # 157953) to a solution that contained 50 mM KH2PO4 (pH 7.0) and oxidized cytochrome c (Sigma # C7752).

All activities are expressed as fold change vs. control group value.

**Respiration measurements**. Rat plantaris muscles were harvested and muscle fibers were permeabilized in a solution containing 10 mM EGTA, 3 mM Mg2+, 20 mM taurine, 0.5 mM DTT, 20 mM imidazole, 0.1 M K+ 2-[N-morpholino]ethane sulfonic acid, pH 7.0, 5 mM ATP, 15 mM phosphocreatine and 50 μg/mL saponin[40]. Mitochondrial oxygen consumption was monitored at 30 ℃ in a 1-mL thermostat-controlled chamber equipped with a Clark oxygen electrode. Respiratory substrates used were: pyruvate 10 mM with malate 10 mM, or succinate 25 mM in presence of rotenone 0.4 μg/mL, or ascorbate 3 mM and N,N,N′,N′-tetramethyl-p-phenylenediamine (TMPD) 0.5 mM. State 3 was obtained by adding 2 mM ADP. After each respiration experiment, the fibers were dried and weighed. Activities are expressed in natom O/min/mg of fiber[38,40].

**Microarray analysis**. RNA profiling in rat plantaris muscle and C2C12 myotubes was performed using a Rat GE 4x44K v3 Microarray Kit and a Mouse GE 4x44K v2 Microarray Kit, respectively (Agilent Technologies, Massy, France). Briefly, 200 ng of total RNA isolated from frozen muscle samples or C2C12 cells was labeled using the Low Input Quick Amp Labeling kit (CY3) (Agilent Technologies, Massy, France), and microarrays were hybridized and scanned following the manufacturer's instructions. Data extraction was performed using Agilent Feature Extraction Software 11.5.1.1. Data quality analysis and processing was performed with bioconductor using Agi4x44PreProcess and Limma packages (Smyth GK, 2004). Background subtraction was performed using the Normexp correction with offset = 50. Quantile normalization was then applied. Features flagged in Feature Extraction as control, non-uniform outliers, saturating or too weak were excluded. Statistical analysis was performed on 26,902 probes for plantaris muscle samples and on 30,094 probes for C2C12 samples with the Limma package. Probes selected for further analysis had a *p*-value < 0.05 for plantaris muscle experiments and a Benjamini–Hochberg adjusted *p*-value < 0.05 for C2C12 experiments. Functional enrichment analysis was performed with g Profiler using a list of downregulated genes differentially expressed in plantaris muscles of vitamin D-depleted old rats and upregulated genes differentially expressed in C2C12 myotubes treated with 10 nM of 1,25(OH)2 vitamin D3. Functional enrichment results were filtered and only the top 10 annotations for Gene Ontology BP, CC, KEGG, and Reactome were kept for functional enrichment visualization using Cytoscape (v3.8) with EnrichmentMap and AutoAnnotate plugins[21,41]. Datasets are available from the GEO database (GSE67274 and GSE64803).

**Statistics and reproducibility**. All data are presented as means ± SEM.

For the human vitamin D supplementation study (interventional study), the sample size was calculated to detect the difference of appendicular skeletal muscle mass at 6 months (primary efficacy endpoint) between the two treatment arms with a two-sided type-I error of 0.05 and a power of 80%.

Statistical analysis for the human interventional study, the long-term vitamin D depletion rat experiment, the *HSA-MCM-VDR*fl/fl transgenic mice study and the cell culture studies with two conditions was performed using a two-tailed unpaired t test. In the long-term vitamin D depletion mouse experiment, the Mann–Whitney *U*-test was used to calculate between-group differences. In cell culture studies comparing more than two group means, one-way analysis of variance (ANOVA) was performed to test the effect of the experimental conditions. When a significant effect was detected, a post hoc Fisher test was applied to locate pairwise differences between conditions. Cell culture data are combined from at least two independent experiments. Linear correlations between variables were evaluated using Pearson's correlation analysis. Statistical analysis was performed using StatView (version 4.02; Abacus Concepts, Berkeley, CA). Significance was set at *p* < 0.05.

**Reporting summary**. Further information on research design is available in the Nature Research Reporting Summary linked to this article.

# Data availability
Microarray analysis data of rat plantaris muscle and C2C12 myotubes have been deposited to the Gene Expression Omnibus (GEO) database (GSE67274 and GSE64803, respectively). Any remaining information can be obtained from the corresponding author upon reasonable request. All requests will need to specify how the data will be used and will require approval by co-investigators. Uncropped and unedited blot/gel images are listed in Supplementary Fig. 4.

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

## Acknowledgements
The authors thank all the members of the animal facilities at the INRAE center in Theix (France) and CKP-Wageningen (The Netherlands) for their valuable assistance in conducting this study.

## Author contributions
Conceptualization: J.S., A.C., M.J., Y.L., M.F., M.V.D., Y.B., and S.W. Acquisition, analysis, or interpretation of data: J.S., A.C., C. Guillet, A.M.M.V., E.M.B.B., C.R., C. Giraudet, V.P., E.M., C.M., P.D., O.L.B., A.B., M.J., Y.L., M.F., M.V.D., N.T., Y.B., and S.W. Data curation: J.S., A.M.M.V., E.M.B.B., C.R., E.M., C.M., M.V.D., and S.W. Funding acquisition: M.J., Y.L., M.F., M.V.D., Y.B., and S.W. Project administration: J.S., A.C., M.J., Y.L., M.F., M.V.D., Y.B., and S.W. Writing—original draft: J.S. and S.W. Writing—review & editing: J.S., A.C., C. Guillet, C.R., C.M., E.M., O.L.B., A.B., M.J., Y.L., M.F., M.V.D., Y.B., and S.W. All authors have read and agreed to the published version of the paper.

## Competing interests
The authors declare the following competing interests: this study was partly supported by a grant from Danone Nutricia Research, Utrecht, The Netherlands, and, M.J., Y.L., M.F., and M.V.D. are employees of Danone Nutricia Research, Utrecht, The Netherlands.

## Additional information

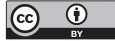

