## [Peer Review File · Communications Biology]

Reviewers' comments:

Reviewer #1 (Remarks to the Author):

Vitamin-D deficiency was induced in rats and changes in muscle function were assessed. It is not clear what was the effect of vitamin-D depletion on serum calcium and phosphorus concentrations and whether these changes in ions may have contributed to the phenotype observed. To attribute all phenotypic changes to alterations in vitamin D metabolite concentrations alone in this in vivo model is an error. This should be addressed. On page 15, line 5, the reviewer notes that the definition of vitamin-D deficiency is a 25 hydroxyvitamin D concentration of less than 25 nM.

That 125-dihydroxyvitamin D alters gene expression in muscle cells is well established. The authors should address how their findings were different from those observed previously. They should explicitly address how their findings could explain phenotypic findings in their model.

Previous studies have demonstrated how vitamin-D metabolites alter molecules responsible for mitochondrial biogenesis. The authors should explicitly discuss what new findings were observed.

The authors are unaware of studies that have used skeletal muscle deletion of the VDR on SM strength (Fogarty et al J Appl Physiol 2021). These studies should be quoted.

Reviewer #2 (Remarks to the Author):

OVERVIEW

The authors show convincing associations between vitamin D status, mitochondrial function and energy expenditure. The potential link of mitochondrial dysfunction to fat accumulation in vitamin D deficiency is interesting. The authors' approach is translational including human, in vivo depletion studies and conditional genetic knockdown of the vitamin D receptor, and in vitro work. Compared with many previously published articles in the area, this manuscript provides more mechanistic insight to the acute and chronic effects of vitamin D deficiency on mitochondrial function and energy expenditure. Overall, I think the manuscript makes an important contribution to the field. I think the authors are over-interpreting the human data, and this must be addressed prior to publication.

METHODS

1. REE- Please be specific about the values that were used within the 1 hour time frame (how time points to report were selected).
2. 25(OH)D- MS is gold standard methodology for assessment of 25(OH)D. Since so many of the conclusions hinge on accurate measurement of 25(OH)D, use of ELISA and radioimmunoassay should be acknowledged as a limitation.
3. Appendicular skeletal muscle mass- the muscle mass data is tenuous. Bioelectrical impedance analysis is an unreliable way to measure regional body composition changes. Computed tomography or magnetic resonance imaging are the most appropriate ways to measure regional body composition changes in humans. BIA is even bad at assessing whole body composition changes when compared with dual-energy X-ray absorptiometry (e.g. PMID: 30351166). BIA estimate of muscle mass may be acceptable to use in surveillance studies including thousands of people, but I don't think the use of BIA is an appropriate measure here. I think all of the statements relating to this outcome should be toned down throughout the manuscript and the limitations of this technique should be explicitly addressed.
4. Strength- similarly, handgrip measures alone provide an incomplete indication of strength and more comprehensive measures would better support clinical relevance of the outcomes. In intervention

studies intended to increase muscle mass and strength, having lower baseline muscle and strength is a known confounder of measuring changes (i.e. smaller people are more likely to show high increases as a % increase of the mass or strength measure). Did the authors confirm that there were no baseline differences in muscle mass and strength between the vitamin D and placebo groups?

5. RNA/DNA- for housekeeping genes, please be specific about the figure legend the reader should refer to.

RESULTS

1. Figure 1.K- The correlation is between 25(OH)D and energy expenditure is in a group of 31 men with ages ranging from 20-79. Since the group is only men, it's inappropriate to make generalizations based on this data. Additionally, the age range is broad and may be driving the correlation between vitamin D and REE. Since younger people have higher energy expenditure, and older people are more likely to be deficient in vitamin D, the authors must demonstrate that this correlation is not driven by the age of the participants (i.e. are the data point on the top right from younger study participants?). It would be great if you could show the individual data points in all of your figures. Please add the participant number for 1.K to the figure legend.

2. Figure 4- Please use consistent capitalization in formatting of categories of DEGs

3. Figure 5- The bottom and sides of some words are cut off. Please double check how these look in the final version of figures (e.g. Tibialis title on top; similar issues in figure 6)

4. Figure 6- The human data in this figure concerns me as described above. Also, vitamin D status in humans is typically conceptualized as adequate or inadequate at a cut point, so changes in 25(OH)D to show biologically meaningful associations was unexpected. Referring to my comments about the BIA measure, I think that the claims relying on this outcome should be more modest. I think percentages on the bottom of page 22 should be removed and expressed as kg changes aligning with the figure. I think "Vitamin D plasma level modulates whole-body energy expenditure in rats and humans" heading page 15 is overstating the strength of the evidence presented and should read "Vitamin D plasma level associates with whole-body energy expenditure in rats and humans."

DISCUSSION

1. I think "The action of vitamin D on REE and muscle mass was verified in older human subjects" is inappropriate because no experiment was conducted, and the participants ranged in age from 20-79.

2. I think "This is the first study to demonstrate a putative role of vitamin D depletion on key mitochondrial function parameters as one of the underlying causes of sarcopenia" is inappropriate because sarcopenia is a human clinical diagnosis and no molecular experiments or measures were conducted in humans. I think the in vivo data indicate that vitamin D deficiency may contribute to sarcopenia.

3. Page 25- "We clearly showed in rodents and in humans that vitamin D contributes to the control of energy expenditure" I don't think any evidence was provided to support a causal or confirmed relationship between vitamin D and energy expenditure. If you can exclude the possibility that age is driving the correlation, then I think an association between 25(OH)D and REE in men is possibly supported.

4. I think a limitations paragraph should be included that addresses, lack of gold standard methodology in human data outcomes, especially deriving regional muscle mass changes from BIA. Also, the REE human observational study was in a wide age range and only included men.

OTHER GENERAL COMMENTS

1. I think the most important issue to address is interpretation of the human data and addressing the limitations.

2. While the many mRNA outcomes are interesting, there is little protein data to support the gene expression conclusions and most of the protein data is from the in vitro studies. It would have been nice to see some western blots confirming the mRNA finding from the in vivo studies. If possible, confirming the gene expression findings with western blot from one of the rodent studies would strengthen the manuscript.

3. In the results section, many "trending" relationships are highlighted. Adding individual data points

to the figures would bring more clarity to these potential relationships and strengthen the manuscript.

4. Please double check manuscript for use of "reduce" where "lower" or similar word should be used e.g. on page 16 "Data obtained over the 24-hour period, showed that energy expenditure was significantly reduced ($p < 0.05$) during the light period ..." EE is significantly lower in one group compared with another group and not reduced from a prior time point. Similarly, using "switch" in "body composition in body composition" at the top of 24 when analysis is between groups.

Point-by-point response to the reviewers' comments:

Reviewer #1 (Remarks to the Author):

Vitamin-D deficiency was induced in rats and changes in muscle function were assessed. It is not clear what was the effect of vitamin-D depletion on serum calcium and phosphorus concentrations and whether these changes in ions may have contributed to the phenotype observed. To attribute all phenotypic changes to alterations in vitamin D metabolite concentrations alone in this in vivo model is an error. This should be addressed.

We agree with the reviewer's statement that vitamin-D depletion could modulate blood calcium and phosphorus concentrations and that ion changes could contributed to the phenotype we observed in rats. To reply to this question, we measured concentrations of several vitamin D-dependent plasma parameters, i.e. calcium, phosphorus and parathyroid hormone (PTH). Finally, we did not notice any significant change in plasma concentrations of calcium, phosphorus and PTH in vitamin-depleted rats compared to control rats (Supplementary Table 2).

We added the Supplementary Table 2 (see below) and added or modified sentences in the Methods section (lines 199-201 page 10), in the Results section (lines 339-340 page 17) and in the discussion section of the revised manuscript (lines 596-597 page 27).

Supplementary Table 2

Vitamin D-dependent parameters in control (n=7) and vitamin D-depleted (n=9) old rats.

	Control	Vitamin D-depleted
Calcium (mg/l)	101.6±1.0	103.5±2.1
Phosphorus (mg/l)	52.05±7.80	43.54±3.44
PTH (pg/ml)	376.4±51.1	347.8±27.7

Sentences added in the Methods section (page 10):

"ELISA kit was also used to determine parathormone (PTH) (Immunotopics, San Clemente, CA, USA). Plasma concentrations of calcium and phosphorus were assessed using a Konelab 20 analyzer (Thermo Electron, Waltham, MA, USA)."

Sentence modified in the Results section (page 17):

"Feeding old Wistar rats on a vitamin D-depleted diet for 9 months resulted in a significant decrease in plasma 25(OH)D concentrations compared to age-matched control animals (-75%; p>0.0001), reaching a severe state of vitamin D deficiency (i.e. <25 nM) (Figure 1a) without affecting plasma levels of calcium, phosphorus and PTH (Supplementary Table 2)."

Sentence added in the discussion section (page 27):

"We did not notice any fluctuation in plasma concentrations of calcium, phosphorus and PTH, so we can hypothesize that the observed effects are specific to vitamin D depletion."

On page 15, line 5, the reviewer notes that the definition of vitamin-D deficiency is a 25 hydroxyvitamin D concentration of less than 25 nM.

We thank the reviewer for the detailed and careful reading of our manuscript. We apologize for this mistake and we corrected it (line 339 page 17 of the revised manuscript).

That 125-dihydroxyvitamin D alters gene expression in muscle cells is well established. The authors should address how their findings were different from those observed previously. They should explicitly address how their findings could explain phenotypic findings in their model.

Previous studies have demonstrated how vitamin-D metabolites alter molecules responsible for mitochondrial biogenesis. The authors should explicitly discussed what new findings were observed. The authors are unaware of studies that have used skeletal muscle deletion of the VDR on SM strength (Fogarty et al J Appl Physiol 2021). These studies should be quoted.

Compared to many previously published papers in the field, our manuscript provides more mechanistic insight into the acute and chronic effects of vitamin D deficiency on mitochondrial function and energy expenditure. Moreover, we used within the same work different models to investigate several mechanisms of action. For example, we used cultured cells with different vitamin D treatments, a mouse model that did not exist until now and omics methods to get a global overview of the effects of this vitamin. The study of a specific and programmed extinction of the VDR at skeletal muscle level, without turning off its gene in the other tissues (which can induce indirect effects on muscle mass and function in the “simple” KO model) made it possible to highlight new results.

We thank the reviewer for reminding us of the article by Fogarty et al. These authors have already published an article using a mouse model similar to ours. However, they performed a study specifically on the diaphragmatic muscle without looking at the muscles of the limbs that play an important role in the effects of vitamin D deficiencies on sarcopenia. These authors did not measure mitochondrial activity or protein turnover and its regulation. Finally, they did not include old animals or samples taken in Humans.

Reviewer #2 (Remarks to the Author):

OVERVIEW

The authors show convincing associations between vitamin D status, mitochondrial function and energy expenditure. The potential link of mitochondrial dysfunction to fat accumulation in vitamin D deficiency is interesting. The authors' approach is translational including human, in vivo depletion studies and conditional genetic knockdown of the vitamin D receptor, and in vitro work. Compared with many previously published articles in the area, this manuscript provides more mechanistic insight to the acute and chronic effects of vitamin D deficiency on mitochondrial function and energy expenditure. Overall, I think the manuscript makes an important contribution to the field.

I think the authors are over-interpreting the human data, and this must be addressed prior to publication.

METHODS

1. REE- Please be specific about the values that were used within the 1 hour time frame (how time points to report were selected).

As suggested by the reviewer and to be more specific in the description of the method we used, we modified sentences as follows (Methods section lines 144-148 page 8 of the revised manuscript):
“Gas samples were collected every minute for one hour. The data of the first 10 minutes were not used because this period corresponds to the stabilization of the metabolism of the subjects under the canopy. The means of the O₂ consumption and CO₂ production values obtained over the last 50 minute period were used in the analysis. REE was calculated using the Weir equation (16).”

2. 25(OH)D- MS is gold standard methodology for assessment of 25(OH)D. Since so many of the conclusions hinge on accurate measurement of 25(OH)D, use of ELISA and radioimmunoassay should be acknowledged as a limitation.

Although the ELISA kits make it possible to measure the plasma concentrations of 25(OH)D more and more accurately and more and more precisely, the reviewer is right, MS is the gold standard analysis for these kinds of measurements. We have not had the opportunity to use such a method but we have analyzed 25(OH)D a large number of times on different types of blood samples. We have never observed any dispersion or inaccuracy in the results obtained. We have added sentences in the discussion section (lines 670-673 page 30) to warn the reader that the methods used are not as accurate as the MS.

3. Appendicular skeletal muscle mass- the muscle mass data is tenuous. Bioelectrical impedance analysis is an unreliable way to measure regional body composition changes. Computed tomography or magnetic resonance imaging are the most appropriate ways to measure regional body composition changes in humans. BIA is even bad at assessing whole body composition changes when compared with dual-energy X-ray absorptiometry (e.g. PMID: 30351166). BIA estimate of muscle mass may be acceptable to use in surveillance studies including thousands of people, but I don't think the use of BIA is an appropriate measure here. I think all of the statements relating to this outcome should be toned down throughout the manuscript and the limitations of this technique should be explicitly addressed.

Again, the reviewer is right about this methodological issue. For reasons of limited access to DEXA, we had to do the measurements by BIA. We have mitigated our interpretations of the BIA data by adding a section describing this methodological limitation in the methods part (lines 167-169 page 9) and in the discussion part (lines 673-679 page 30) of the article.

4. Strength- similarly, handgrip measures along provide an incomplete indication of strength and more comprehensive measures would better support clinical relevance of the outcomes. In intervention studies intended to increase muscle mass and strength, having lower baseline muscle and strength is a known confounder of measuring changes (i.e. smaller people are more likely to show high increases as a % increase of the mass or strength measure). Did the authors confirm that there were no baseline differences in muscle mass and strength between the vitamin D and placebo groups?

We added these information in Results section as follows (lines 537-541 page 25 of the revised manuscript) :
“At the beginning of the supplementation, the appendicular skeletal muscle mass was significantly lower in the placebo group compared with the vitamin D-supplemented group (16.8±0.5 Kg vs. 21.6±0.8 Kg, respectively), while no significant difference was observed for the grip strength (20.6±0.7 Kg vs. 21.4±0.7 Kg, respectively).” As the reviewer says, increases in muscle mass and strength are greatest in smaller, frailer people. In our work, muscle mass was lower in subjects selected for the

placebo group. Despite this, their muscle mass and strength were not modified during the study period, unlike subjects receiving vitamin D in whom these parameters increased.

5. RNA/DNA- for housekeeping genes, please be specific about the figure legend the reader should refer to.

To be more specific, we added these information as follows (Methods section lines 276-278 page 13 of the revised manuscript).

“mRNA levels were normalized to the corresponding housekeeping genes depending on species as indicated in legends of Tables 1 and 2, Figures 3 and 5, and Supplementary Figure 3.”

RESULTS

1. *Figure 1.K- The correlation is between 25(OH)D and energy expenditure is in a group of 31 men with ages ranging from 20-79. Since the group is only men, it's inappropriate to make generalizations based on this data. Additionally, the age range is broad and may be driving the correlation between vitamin D and REE. Since younger people have higher energy expenditure, and older people are more likely to be deficient in vitamin D, the authors must demonstrate that this correlation is not driven by the age of the participants (i.e. are the data point on the top right from younger study participants?). It would be great if you could show the individual data points in all of your figures. Please add the participant number for 1.K to the figure legend.*

We agree with the reviewer's opinion. Our generalizations based on our study data are inappropriate. We modified sentences as follows (Results section lines 379-380 page 18 of the revised manuscript) :

“Taken together, our data collected in rats and in a group of healthy men points to a tight relationship between plasma vitamin D levels and REE.”

We have also modified some part of sentences in the discussion section.

We agree with the reviewer's opinion. Usually, “younger people have higher energy expenditure, and older people are more likely to be deficient in vitamin D”. As suggested, to show that the correlation between REE adjusted to lean body mass and plasma vitamin D concentrations is not driven by the age of the participants, we reported data from young subjects and from old subjects in the figure 1k. As shown in these figure (see below), the reviewer is right: the data point on the top right corresponds to a younger participant but we can also observe that the 3 other participants that were characterized by the highest plasma vitamin D concentrations were old subjects. Furthermore, as expected REE adjusted to lean body mass is significantly higher in young participants than in old participants (31 ± 1 vs. 28 ± 1 kcal/day/Kg, respectively) but we found no between-group difference in the plasma vitamin D concentration (11.8 ± 2.1 nM in young vs. 12.7 ± 1.8 nM in old subjects). According to these results, we think that the correlation we observed between REE adjusted to lean body mass and plasma vitamin D concentrations is not driven by the age of the study participants.

As suggested by the reviewer, we also reported all data points in the figures excepted for figures 6c, 6d and 6e because the number of data points is too large (115).

To be more specific in the description of the group of 31 men, we modified sentences as follows (Methods section lines 137-139 page 8 of the revised manuscript) :

“Thirty-one healthy male subjects, including 15 young (from 20 to 35 years old) and 16 older individuals (over 60 years old), with a Body Mass Index (BMI) between 21.4 and 34.2 kg/m² were included in the study, as previously reported (14, 15).”

We thank the reviewer for pointing out this issue. We added the participant number and some information in the legend of the figure 1k as follows (Results section lines 801-803 page 34 of the revised manuscript) :

“(K) Correlations between resting energy expenditure (REE) adjusted to lean body mass and plasma 25(OH)D concentrations in 31 male subjects including 15 young (from 20 to 35 years old) and 16 older individuals (over 60 years old) (r = Pearson’s correlation coefficient).”

2. Figure 4- Please use consistent capitalization in formatting of categories of DEGs

We apologize for not being specific enough. To generate Figure 4 we used a suit of applications in Cytoscape. Using EnrichmentMap and AutoAnnotate we were able to clusterise related functional annotations (annotations of these functions are all in capitalized letters) and adding an annotation of these clusters (lower case annotations). The size of the cluster’s annotation font relies on the size of the cluster generated (the higher the number of nodes is the bigger the font is).

We added information in the legend of the figure 4 as follows (lines 846-850 page 36):

“Using the Cytoscape applications, EnrichmentMap and AutoAnnotate, related functional annotations (annotations of these functions are all in capitalized letters) were clusterised and an annotation of these clusters (lower case annotations) was added. The size of the cluster’s annotation font relies on the size of the cluster generated (the higher the number of nodes is the bigger the font is).”

3. Figure 5- The bottom and sides of some words are cut off. Please double check how these look in the final version of figures (e.g. Tibialis title on top; similar issues in figure 6)

We thank the reviewer for the detailed and careful reading of our manuscript. We apologize for these layout errors and we corrected them (pages 45 and 46 of the revised manuscript).

4. Figure 6- The human data in this figure concerns me as described above. Also, vitamin D status in humans is typically conceptualized as adequate or inadequate at a cut point, so changes in 25(OH)D to show biologically meaningful associations was unexpected. Referring to my comments about the BIA measure, I think that the claims relying on this outcome should be more modest. I think percentages on the bottom of page 22 should be removed and expressed as kg changes aligning with the figure. I think "Vitamin D plasma level modulates whole-body energy expenditure in rats and humans" heading page 15 is overstating the strength of the evidence presented and should read "Vitamin D plasma level associates with whole-body energy expenditure in rats and humans."

In the human interventional study, we recruited 115 elderly subjects who were vitamin D deficient and presarcopenic. We included male and female, and, normal weight and overweight participants. We apologize because we made a mistake in the first manuscript. We presented results from only the normal weight participant group (n=59). In the revised manuscript, we modified the figures 6c, 6d, 6e and 6f which present results from the 115 participants (normal weight + overweight) of the interventional study as follows:

Figure 6: Vitamin D status modulates muscle mass and function in old mice and in vitamin D-deficient older subjects.

As suggested by the reviewer, we removed the percentages and modified the sentence as follows (lines 541-547 page 25):

“Subjects who received 10,000 IU of cholecalciferol three times per week for 6 months exhibited a significantly greater increase in plasma 25(OH) D from baseline compared with placebo-group participants (Figure 6c). Furthermore, mean changes in appendicular skeletal muscle mass and handgrip strength from baseline to 6 months were very significantly higher in the vitamin D-supplemented group than in the placebo group (+0.57±0.09 Kg vs. +0.07±0.05 kg for appendicular skeletal muscle mass gain and +0.85±0.31 Kg vs. +0.09±0.20 Kg for handgrip strength gain) (Figure 6d and 6e).”

We agree that “Vitamin D plasma level modulates whole-body energy expenditure in rats and humans” at the end of page 17 is overstating the strength of the evidence presented after. So as suggested, we replaced the sentence by “Vitamin D plasma level associates with whole-body energy expenditure in rats and humans.” (line 357)

DISCUSSION

1. I think “The action of vitamin D on REE and muscle mass was verified in older human subjects” is inappropriate because no experiment was conducted, and the participants ranged in age from 20-79.

The discussion section has been changed as appropriate (lines 558-560).

“Although, it did not give any cause-effect relationship, a correlation between vitamin D fluctuation and metabolism and REE and muscle mass changes was observed in older human subjects and muscle-specific VDR KO mice”.

2. I think “This is the first study to demonstrate a putative role of vitamin D depletion on key mitochondrial function parameters as one of the underlying causes of sarcopenia” is inappropriate because sarcopenia is a human clinical diagnosis and no molecular experiments or measures were conducted in humans. I think the in vivo data indicate that vitamin D deficiency may contribute to sarcopenia.

The discussion section has been changed as appropriate (lines 567-571).

“In addition, the present study demonstrated a putative role of vitamin D depletion on key mitochondrial function parameters in skeletal muscle. Some previous data conducted in vitamin D deficiency in mice suggest that vitamin D-mediated regulation of mitochondrial function may underlie the exacerbated muscle fatigue and performance deficits observed during vitamin D deficiency.”

3. Page 25- “We clearly showed in rodents and in humans that vitamin D contributes to the control of energy expenditure” I don’t think any evidence was provided to support a causal or confirmed relationship between vitamin D and energy expenditure. If you can exclude the possibility that age is driving the correlation, then I think an association between 25(OH)D and REE in men is possibly supported.

The discussion section has been changed as appropriate (lines 600-602).

“We clearly showed in rodents and in men that vitamin D fluctuations are associated with the control of energy expenditure.”

4. I think a limitations paragraph should be included that addresses, lack of gold standard methodology in human data outcomes, especially deriving regional muscle mass changes from BIA. Also, the REE human observational study was in a wide age range and only included men.

A paragraph with technical limitations has been added to the discussion section (lines 670-679).

OTHER GENERAL COMMENTS

1. I think the most important issue to address is interpretation of the human data and addressing the limitations.

We have mitigated the interpretation of human data and we have added a limitation paragraph (lines 670-679).

2. While the many mRNA outcomes are interesting, there is little protein data to support the gene expression conclusions and most of the protein data is from the in vitro studies. It would have been nice to see some western blots confirming the mRNA finding from the in vivo studies. If possible, confirming the gene expression findings with western blot from one of the rodent studies would strengthen the manuscript.

We agree with the reviewer's criticism. To strengthen the manuscript, we added protein data from one of the rodent studies. We quantified protein levels of NRF1, NRF2 and mitochondrial complex 4 subunit 4 in gastrocnemius muscles of tamoxifen-treated HSA-MCM-VDRfl/fl mice and corn oil-treated HSA-MCM-VDRfl/fl mice (controls) by western-blot. Results are presented in Figures 5h and 5i of the revised manuscript and support the gene expression conclusions for NRF1 and NRF2.

We added information about antibody references in Methods section of the revised manuscript (lines 262-264).

We modified Figure 5 legend as follows (lines 865-867 page 37 of the revised manuscript):
"Immunoblots and red Ponceau staining (h), and immunoblot quantifications (i) of NRF1, NRF2 and complex IV subunit 4 proteins in gastrocnemius muscle of tamoxifen-treated HSA-MCM-VDRfl/fl mice (n=5) and corn oil vehicle-treated HSA-MCM-VDRfl/fl mice (n=5) (Mouse experiment 2)."

We added sentences in the Results section as follows (lines 520-524 page 24 of the revised manuscript):

"Quantification of protein levels confirmed gene expression data and measurements of mitochondrial complex activities. Gastrocnemius muscle NRF1 and NRF2 protein contents were significantly lower in mVDR KO mice than in vehicle-treated mice (-20% and -35%, respectively). In addition, muscle VDR deficiency reduced muscle complex IV subunit 4 protein content (-22%, p<0.05) (Figures 5h and 5i)."

3. In the results section, many “trending” relationships are highlighted. Adding individual data points to the figures would bring more clarity to these potential relationships and strengthen the manuscript.

As suggested by the reviewer and as indicated in the style and formatting guide of Communications biology journal, we added all data points to the figures excepted for figures 6c, 6d and 6e because the number of data points is too large (115).

4. Please double check manuscript for use of “reduce” where “lower” or similar word should be used e.g. on page 16 “Data obtained over the 24-hour period, showed that energy expenditure was significantly reduced

(p

As suggested by the reviewer, we modified sentences in the Results section as follow:

Lines 350-351 Page 17:

“Note that the weight of plantaris muscle, a type II fibre-prominent muscle, tended to be lower in vitamin D-depleted rats than in control rats (-18%, $p=0.05$) (Figure 1d).”

Lines 368-371 Page 18:

“Data obtained over the 24-hour period, showed that energy expenditure was significantly lower during the light period and tended to be lower during the dark period in vitamin D-depleted rats than in their control counterparts (Figures 1i and 1j).”

Lines 392-394 Page 19

“More specifically, the state 3 respiration rate was lower by 20%, 24% and 30% with pyruvate/malate, succinate and ascorbate/TMPD, respectively, in the vitamin D-depleted group compared with control rats (Figure 2a).”

REVIEWERS' COMMENTS:

Reviewer #2 (Remarks to the Author):

Thank you for your thorough response to my review. I think you have completely addressed my concerns.